# A modular implementation of an effective interaction approach for harmonically trapped fermions in 1D

Lukas Rammelmüller[1,2*], David Huber[3], Artem G. Volosniev[4†]

**1** Arnold Sommerfeld Center for Theoretical Physics (ASC), University of Munich, Theresienstr. 37, 80333 München, Germany
**2** Munich Center for Quantum Science and Technology (MCQST), Schellingstr. 4, 80799 München, Germany
**3** Institut für Kernphysik (Theoriezentrum), Technische Universität Darmstadt, Schlossgartenstraße 2, D-64289 Darmstadt, Germany
**4** IST Austria (Institute of Science and Technology Austria), Am Campus 1, 3400 Klosterneuburg, Austria
* lukas.rammelmueller@physik.uni-muenchen.de
† artem.volosniev@ist.ac.at

November 8, 2022

## Abstract

We introduce a generic and accessible implementation of an exact diagonalization method for studying few-fermion models. Our aim is to provide a testbed for the newcomers to the field as well as a stepping stone for trying out novel optimizations and approximations. This userguide consists of a description of the algorithm, and several examples in varying orders of sophistication. In particular, we exemplify our routine using an effective-interaction approach that fixes the low-energy physics. We benchmark this approach against the existing data, and show that it is able to deliver state-of-the-art numerical results at a significantly reduced computational cost.

# 1  Introduction

Analytical or numerical solutions to few-body models are often key elements behind our understanding of properties of cold-atom systems (see Ref. [1] and references therein). In particular, few-body physics underlies many important phenomena in quasi-one-dimensional setups [2,3]. The predominant method of choice to solve few-body problems is exact diagonalization (ED) in a truncated many-body Hilbert space, also known as the full-configuration interaction (FCI) method [4]. The ED approach is arguably the most straightforward numerical method to obtain accurate results in a relatively broad parameter space.

However, achieving the convergence in ED calculations might be a daunting task even for small systems in one spatial dimension (1D), due to its poor convergence for increasing particle number. To (partly) overcome prohibitive cost of numerically exact values, several approaches have been proposed in the literature. Some of those approaches merely employ different truncation schemes of the Hilbert space based upon, for example, enforcing an energy restriction of the many-body basis states [5], freezing irrelevant degrees of freedom [6] or restricting the excitations to be of a certain kind [7]. Other developments include: optimized choices of certain numerical or physical parameters [8], similarity transformed Hamiltonians [9, 10] and interactions [11–13]. Despite the differences in these various optimizations, the setup of the corresponding ED programs is fairly generic, allowing for an abstract implementation that covers all of those cases.

In the present paper, we discuss such an implementation, and illustrate it with a focus on the effective-interaction approach that relies on a particularly efficient renormalization of the interaction. Let us briefly motivate the usage of the effective-interaction approach. A typical problem that one faces while diagonalizing a Hamiltonian in a restricted Hilbert space is finding the relation between the couplings of the truncated theory and the ones in the original theory, see, e.g., the discussion in [14]. It can be solved by considering the two-body problem in a restricted space and matching its ground-state energy to the exact value, if available. For harmonically trapped fermions with zero-range interactions, this can be done using the known solutions of Refs. [15, 16]. The effective-interaction approach extends this idea by matching the low-lying two-body energy spectrum in a restricted Hilbert space to the exact one. This is achieved by tuning the entire interaction matrix (i.e., two-body coefficients in second quantization) instead of only the couplings in the Hamiltonian. Technically, it constitutes an uncontrolled approximation when applied to problems with more than two particles. However, for trapped few-fermion systems, one finds that the use of such an approach dramatically improves the convergence properties of an FCI calculation. Thereby, systems with larger particle numbers and/or stronger interactions become feasible for investigation, see, e.g., Ref. [17].

The aim of this paper is twofold: Firstly, we aim to introduce a simple and lightweight implementation of a generic approach to FCI calculations in truncated Hilbert spaces, which is freely available as the Julia package `FermiFCI.jl` [18]. Our implemenation takes away the inital barrier of writing generic pieces of code (so-called boilerplate code) and allows one to quickly test new ideas for approximations or optimizations. The key focus behind the design of `FermiFCI.jl` is on its usability and easy access, which we hope will be suitable for both theorists and experimentalists. To illustrate the versatitlity of our implementation, we shall discuss a few examples. Secondly, we want to provide an overview of the FCI method with an effective interaction in the context of 1D harmonically trapped few-fermion problems and discuss its convergence properties. To this end, we detail a general idea of the approach and show how it can be implemented with the provided Julia package.

The remainder of this paper is organized as follows: In Sect. 2 we briefly introduce the relevant model(s) and our notation along with a concise overview of ED in truncated Hilbert spaces. Furthermore, we introduce `FermiFCI.jl` and point out how to leverage out a freely accessible Julia implementation [18]. In Sect. 3 we discuss three physical examples to outline the modularity and ease-of-use of the package. In Sect. 4 we provide an example of the usage of the provided code package for the effective two-body interaction method. In this example, we provide an extensive benchmark against the available data. Finally, a brief outlook is presented in Sect. 5. The source code that includes the discussed examples is freely available [19].

## 2   Exact diagonalization of fermionic models

In this section, we introduce our notation and provide a concise overview on the numerical treatement of a few fermions interacting via an arbitrary two-body potential in confining potentials of any shape. For an in-depth discussion of exact diagonalization methods in this context see, e.g., Ref. [4].

## 2.1 Hamiltonian and notation

We are interested in diagonalizing Hamiltonians that describe two-component systems with an arbitrary two-body interaction. In second quantization, such Hamiltonians may be written as

$$
\begin{aligned}
\hat{H} &= \hat{H}_1 + \hat{H}_2 \\
&= \sum_\mu \sum_{i,j} C_{ij}^\mu \, \hat{\psi}_{\mu,i}^\dagger \hat{\psi}_{\mu,j} + \sum_{\mu\nu} \sum_{ijkl} V_{ijkl}^{\mu\nu} \, \hat{\psi}_{\mu,i}^\dagger \hat{\psi}_{\nu,j}^\dagger \hat{\psi}_{\nu,l} \hat{\psi}_{\mu,k},
\end{aligned}
\tag{1}
$$

where $\hat{\psi}_{\mu,k}^\dagger$ and $\hat{\psi}_{\mu,k}$ create and annihilate a fermion of type $\mu$ in the $k$-th single-particle orbital (SPO). They obey the standard fermionic anticommutation relations: $\{\hat{\psi}_{\mu,i}, \hat{\psi}_{\mu,k}^\dagger\} = \delta_{i,k}$; $\{\hat{\psi}_{\mu,i}^\dagger, \hat{\psi}_{\mu,k}^\dagger\} = \{\hat{\psi}_{\mu,i}, \hat{\psi}_{\mu,k}\} = 0$. As written, Eq. (1) describes arbitrarily many flavors of fermions. However, the following discussion focuses on the case with only two species, most commonly referred to as spin-up and -down, i.e., $\mu, \nu \in \{\uparrow, \downarrow\}$.

The specific physical content of Eq. (1) is hidden in the one- and two-body coefficients $C_{ij}^\mu$ and $V_{ijkl}^{\mu\nu}$, given by

$$
C_{ij}^\mu = \langle \phi_i^\mu | \hat{C} | \phi_j^\mu \rangle = \int \mathrm{d}x \, \mathrm{d}y \, \langle \phi_i^\mu | x \rangle \langle x | \hat{C} | y \rangle \langle y | \phi_j^\mu \rangle
\tag{2}
$$

$$
= \int \mathrm{d}x \, \mathrm{d}y \, \phi_i^{\mu *}(x) C(x, y) \phi_j^\mu(y),
\tag{3}
$$

and

$$
V_{ijkl}^{\mu\nu} = \langle \phi_i^\mu \phi_j^\nu | \hat{V} | \phi_k^\mu \phi_l^\nu \rangle
\tag{4}
$$

$$
= \int \mathrm{d}x \, \mathrm{d}y \, \phi_i^{\mu *}(x) \phi_j^{\nu *}(y) \, V(x, y) \, \phi_k^\mu(y) \phi_l^\nu(x),
\tag{5}
$$

where $|\phi_i^\mu\rangle$ represents a SPO and $\phi_i^\mu(x)$ is its spatial representation, which is employed to solve the problem of our interest.

Often, it is convenient to work in a basis of single-particle orbitals where the one-body term is diagonal, i.e., where we can write

$$
\langle \phi_i^\mu | \hat{C} | \phi_j^\mu \rangle = C^\mu(x) \, \delta_{ij}.
\tag{6}
$$

A well known example, as we shall see below, are the eigenfunctions of the harmonic oscillator, which allow for an efficient numerical treatment and even closed-form expressions for the coefficients $V_{ijkl}^{\mu\nu}$, at least for problems where the particles reside in a harmonic trap and interact via a delta-function potential. However, for many problems the one-body term may only be diagonalized numerically. Therefore, we do not assume the validity of Eq. (6) in general. Although we do not discuss it in the present work, `FermiFCI.jl` can be used with general one-body terms, as we illustrate for a few-fermion system with a magnetic impurity in Ref. [20].

## 2.2 Diagonalization in a truncated basis

Here, we briefly elaborate on the numerical diagonalization of the generic model given in Eq. (1). The main point of any diagonalization scheme is to render the problem finite, i.e.,

one must truncate the infinite sums that appear in Eq. (1). One of the most straightforward (and most used) truncation methods would be to simply cut all sums over the single-particle indicies at some value $n_b$ and discard all states where higher-lying orbitals are occupied. We shall refer to this truncation as the *"plain cutoff"*.

Regardless of how truncation is performed, one obtains a finite-dimensional Hilbert space, $\mathcal{H}$, which contains all permissible many-body states, i.e., the many-body basis where the problem is defined. The states can be described via their occupation numbers in the chosen SPOs, where a certain (infinite) subset of physical states has been omitted. Once a suitable many-body basis has been established, one must construct the elements of the Hamiltonian of interest. These are commonly stored in the form of a sparse matrix, since most of the matrix elements for physical problems actually vanish. The next step is diagonalization of this matrix. In practice, the implicitly restarted Arnoldi Method (predominantly the ARPACK implementation [21]) is the method of choice to extract the lowest part of the energy spectrum for large matrices.

Of course, omitting physical states inevitably introduces an error with respect to the exact values. The hope is, however, that the discarded states are unessential for the low-energy physics. To confirm this expectation, one should be able to systematically improve the accuracy by considering a sequence of approximations with successively larger $\mathcal{H}$ and either use a "large enough" many-body basis or extrapolate to the infinite-basis limit. A caveat, particularly for more than only a few particles, is the prohibitive scaling of the dimensionality of $\mathcal{H}$. For example, the size of the Hilbert space with the plain cutoff reads

$$\dim \mathcal{H} = \binom{n_b}{N_\uparrow}\binom{n_b}{N_\downarrow}, \tag{7}$$

where the $N_\sigma$ denotes the number of particles for a given flavor. The size of the many-body basis grows rapidly with $n_b$ and $N_\sigma$[1], quickly leading to matrices whose diagonalization is beyond what is feasible even on the largest supercomputers. For example, a configuration with $N_\uparrow = 3, N_\downarrow = 3$ and $n_b = 30$ (which we will see below is usually far from convergence without any further improvements) results in $\dim \mathcal{H} \approx 1.5 \times 10^7$, which is at the limit of current high-perfomance computing (HPC) platforms.

Despite the fact that hardware requirements for performing diagonalization of large matrices might be high, the ED approach is useful for many cases of interest. The necessity and/or quality of an extrapolation to $n_b \to \infty$ strongly depends on the parameters of the model. Often, the strength of the particle-particle interaction determines whether the ED approach will lead to accurate results.

## 2.3   Structure and usage of `FermiFCI.jl`

As we mentioned in the introduction, there are several ideas on how to (partly) overcome the necessity to work with large many-body bases in ED approaches. The setup of any of those ideas is fairly generic, and it is our goal here to provide a lightweight and versatile package that will allow one test those ideas without a need to re-implement the standard sections of code. To this end, we introduce `FermiFCI.jl` [18], a simple and generic solver of few-fermion problems written in the Julia language. It provides a modular and easy-to-use tool

---

[1]Typically, $n_b \gg N_\sigma$ leading to a polynomial growth of the Hilbert space as a function of $n_b$ for a fixed number of particles, $\dim \mathcal{H} \simeq n_b^{N_\uparrow + N_\downarrow}/(N_\uparrow! N_\downarrow!)$.

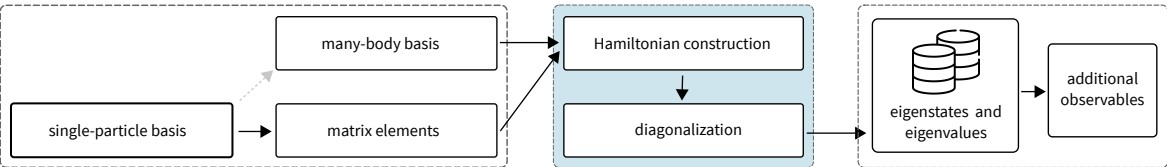

Figure 1: **Flow diagram of a generic exact diagonalization implementation.** Elements in the leftmost block describe the input which depends on the system (model etc.), the central block reflects generic core functionality of `FermiFCI.jl` (independent of the physical model of interest) and the rightmost block summarizes results of the calculations.

for benchmarking new approaches. While the code is performant and allows one to obtain state-of-the-art numerical values, the implementation decidedly was not over-optimized – we rather focused on readability and usability. This subsection provide information on how to use `FermiFCI.jl` , on the setup of the code and some general design principles. The next section illustrates the framework using three examples.

A generic flow chart of `FermiFCI.jl` (and most other diagonalization schemes) is depicted in Fig. 1, where the ED approach is split into three parts. The parts of the calculation that depend on the physical model under investigation (the single-particle basis, the many-body basis and the resulting matrix elements) are indicated in the left box. The core functionality (blue box), namely the setup of the Hamiltonian matrix as well as its diagonalization, is independent of the physical model of interest. So is the further computation of observables, which mainly requires the wavefunctions obtained during the diagonalization of $\hat{H}$. Below, we briefly describe the parts depicted in Fig. 1, and the usage and conventions for `FermiFCI.jl` . Additional information is provided together with the documentation of the code [18].

### 2.3.1 Physical input

As indicated in the left part of Fig. 1, there are three ingredients that define the physics of interest: the many-body basis (essentially a list of permissible many-body configurations), the single-particle orbitals, and the matrix elements $C_{ij}^{\mu}$ and $V_{ijkl}^{\mu\nu}$, see Eq. (1).

**Single-particle orbitals** are one of the basic building blocks in `FermiFCI.jl` . They are used to construct model coefficients as well as in the computation of further ground-state observables such as density profiles. The SPOs should be represented as data types, which is a common design choice in Julia packages. The functionality of the type itself may be implemented rather freely, there merely are two requirements: Firstly, when a type is called with an integer $n$ the single-particle energy of the $n$-th orbital has be returned. Secondly, when called with an integer $n$ and some coordinate $x$ (with the appropriate dimension), the value of the function of the $n$-th orbtial at the position $x$ is returned. The following listing contains an example implementation for the SPOs of the 1D harmonic oscillator (cf. Eq. (10)), which will be used for some of the examples in Sec. 3.

```
1    using SpecialPolynomials
2
3    # Specific 1D orbital.
4    struct HOOrbital1D <: Orbital
5      m :: Float64 # Mass.
```

```
 6        omega :: Float64 # Trap frequency.
 7    end
 8
 9    # Callable that returns the energy.
10    (orb::HOOrbital1D)(i::Integer) = begin
11        return (i-0.5)*orb.omega
12    end
13
14    # Callable that returns the spatial wavefunction.
15    (orb::HOOrbital1D)(i::Integer, x::Number) = begin
16        n = big(i-1) # Formulas are typically zero-based, Julia is one-based.
17        xi_inv = sqrt(orb.m*orb.omega) # Inverse of HO length.
18        return Float64(
19                    (xi_inv^2/pi)^0.25 / sqrt(2.0^n *
20                    factorial(n)) * exp(-0.5*orb.(x*xi_inv)^2) *
21                    basis(Hermite, Int64(n))(x*xi_inv)
22                 )
23    end
```

This piece of the code reflects the common choice that potential scale parameters (here the mass $m$ and the frequency $\omega$; $\hbar$ is always set to 1) are stored as type members whereas actual functionality (such as the energy or the spatial representation) is relegated to calls on a given instance of the orbital type. The structure is used by instantiating a type with a given scale parameter:

```
 1    # Instantiate a set of 1D HO orbitals with scale parameters set to unity.
 2    ho_orbital = HOOrbital1D(1.0, 1.0)
```

**Matrix elements**  contain all details of the relevant potential and kinetic energies. For the core functionality of `FermiFCI.jl` it does not matter how the matrix coefficients are obtained, as long as they are provided as matrices and rank-4 tensors for one- and two-body coefficients, respectively. For example, the one-body coefficients for the harmonic oscillator (without any additional single-particle terms, cf. Sect. 3.1) may be constructed as follows:

```
 1    # Use list comprehension to construct the one-body coefficients.
 2    c_ij = [i==j ? ho_orbital(i) : 0 for i=1:nb,j=1:nb]
```

Here, we have exploited the fact that the single-particle coefficients are diagonal in the HO basis for particles in a harmonic trap. We note, that the construction of $C_{ij}$ could have as well been done without the one-body orbitals by simply using the explicit formula $C_{ij} = \delta_{ij}\hbar\omega(i+1/2)$. However, the use of types for the SPOs is recommended as it improves readability of the code and reduces the risk of errors. For the construction of two-body coefficients we refer to the examples below as well as to the corresponding codes [19].

**The many-body basis**  is essentially a list of states that represent a given particle configuration in the Fock basis. Generically, a basis-state $\psi$ can be split into two components, one for each spin flavor, via a direct product:

$$|\psi\rangle = |\psi_\uparrow\rangle \otimes |\psi_\downarrow\rangle. \tag{8}$$

For concreteness, consider a state of $2 \uparrow +1 \downarrow$ particles which reside in orbitals $n = 1, 3$ and $n = 5$, respectively. This state may be written as $|\psi\rangle = |00101\rangle \otimes |10000\rangle$ in the Fock representation. Here, the occupation number of the lowest orbital is assumed to be on the right such that these basis states may be efficiently represented as binary numbers.

Although, it is natural to represent this type of states as a bitstring (i.e., integer values), it is useful to keep the implementation abstract. To this end, `FermiFCI.jl` provides two datatypes: Objects that represent the combined basis state $|\psi\rangle$ are of the type `FullState` whereas the spin-states $|\psi_\sigma\rangle$ are represented by the type `SpinState`. It is not strictly necessary, but it is recommended that these pre-defined types are used whenever basis states are handled for reasons of type safety and readability. Moreover, several functions such as creation, anihilation and indexing are defined on these types. Finally, functions that convert between the two types are also provided:

```
1    # Combine two spin states to a full state.
2    psi = s_to_f(psi_up, psi_down)
3
4    # Convert from a full state back to separate spin states.
5    psi_up, psi_down = f_to_s(psi)
```

With the nature of the states layed out, the only thing left to do is the construction of the many-body basis, which should be a one-dimensional array of the type `FullState`. This task is left to the user. However, some standard examples (see below) are provided in the package `FermiFCI.Utils`.

### 2.3.2 Core functionality

The diagonalization of a given Hamiltonian may be viewed as a two-step process: First, we compute the matrix elements of the Hamiltonian in the specified many-body basis by using the `construct_hamiltonian` method (see App. A for details on the implementation). This function returns an object of the type `Hamiltonian` which is essentially a sparse matrix[2]. For a problem where interactions occur only between $\uparrow$- and $\downarrow$-particles, which is natural in cold-atom systems, we may construct the Hamiltonain matrix as follows:

```
1    using FermiFCI
2
3    hamiltonian = construct_hamiltonian(
4      mb_basis,
5      up_coeffs=c_ij,
6      down_coeffs=c_ij,
7      up_down_coeffs=v_ijkl,
8    )
```

Here, the coefficients `v_ijkl` reflect, for instance, the coefficients in Eq. (11) for the interaction between $\uparrow$ and $\downarrow$ particles. Having obtained the elements of the matrix we simply extract the low-energy spectrum and eigenvalues via diagonalization, for which we provide the convenient wrapper `diagonalize`:

---

[2]For larger calculations it is advisable to compute the elements of the Hamiltonian on the fly while performing the matrix-vector operations, since the memory requirements will be steep. For the sake of simplicity and readability, we opted for the storage of the matrix.

```
1    eigenvalues, eigenstates = diagonalize(hamiltonian, param)
```

By default this returns the lowest few eigenvalues (sorted) with the associated eigenstates as a 2D array. As explained more extensively in the documentation of the code, the dictionary `param` holds all relevant computational and physical parameters which includes the settings for the diagonalization routines (number of eigenvalues, maximal iterations, etc.). Specifically, the diagonalization can be toggled between Lanczos/Arnoldi (default) and full diagonalization with the same behavior of the output. For details we refer to the code of the provided examples [19].

### 2.3.3 Output and further observables

Once the diagonalization is done, one might be interested in quantities other than the low-energy spectrum. To calculate them, one should use the produced eigenstates. In general, the implementation of these further observables is left to the user. However, for convenience some common observables, such as density profiles as well as one- and two-body density matrices, are provided in the package `FermiFCI.Utils`, see the code examples for details.

## 3   Three examples

Having laid out the general structure of the code, we show the application of the framework to two specific problems, namely a few fermions in a harmonic confinement (examples I and II) as well as a few mass-imbalanced fermions in a flat trap (example III). The source code for these examples is included in the repository [19].

### 3.1   Example I: 1D harmonically trapped fermions with plain cutoff

For our first demonstration of `FermiFCI.jl` we consider an ensemble of harmonically trapped two-component fermions in one spatial dimension whose Hamiltonian in first quantization reads

$$\hat{H} = \sum_{i=1}^{N} \left( -\frac{\hbar^2}{2m} \nabla_{x_i}^2 + \frac{m\omega^2}{2} x_i^2 \right) + \frac{g}{2} \sum_{i \neq j} \delta(x_i - x_j) \tag{9}$$

where $N$ denotes the total particle number, $\omega$ is the frequency of the trap, $m$ is the mass of a particle. The strength of the interaction is given by $g$. The characteristic length scale of the problem is given by the harmonic oscillator length $\xi = \sqrt{\hbar/m\omega}$. The model in Eq. (9) is one of the most studied few-body problems due to recent experiments with quasi-one-dimensional systems, see, e.g., [22–24]. Accurate benchmark data for this model is available in a number of papers, see, e.g., [25, 26], which makes it well suited for the purpose of this section.

The natural choice for the single-particle orbitals $|\phi_k^\mu\rangle$ are the eigenstates of the harmonic oscillator. In 1D [3] , the coordinate representation of the $n$-th eigenfunction, corresponding to

---

[3]Note that for 2D and 3D problems it may be more convenient to work in another eigenbases (e.g., in polar or spherical coordinates) that may allow for a better implementation of certain symmetries. `FermiFCI.jl` is useful also for those cases, one merely must specify a unique mapping between the linear counter of the state in the single-particle basis and all quantum numbers of that state.

the energy $\varepsilon_n = \hbar\omega(n + \frac{1}{2})$ reads

$$\phi_n^\mu(x) = \frac{1}{\sqrt{2^n n!}} \left(\frac{1}{\pi\xi^2}\right)^{\frac{1}{4}} e^{-\frac{1}{2}\left(\frac{x}{\xi}\right)^2} H_n(\tfrac{x}{\xi}), \tag{10}$$

where $H_n(\frac{x}{\xi})$ denotes the $n$-th Hermite polynomial. By construction, the lowest state corresponds to $n = 0$. For the remainder of this work we will set $\xi = 1$ and present all energies in dimensionless units of $\hbar\omega$ when working with harmonically trapped fermions.

As already pointed out above, for the simplest implementation, namely a plain single-particle basis cutoff, these are all required ingredients to start computing with the present framework. To this end, we simply use the previously defined datatype for the single-particle orbitals of the 1D harmonic oscillator. Piecing together the various functions provided by the `FermiFCI.jl` framework one may in principle compute all expectation values of interest out of the box (this is the approach for particles in a flat trap discussed in Sect. 3.3). However, we may save some effort by writing the interaction coefficients $V_{ijkl}^{\mu\nu}$ in a more convenient way, thereby skipping the requirement of potentially unstable numerical integration. As this will be useful for several examples to follow, we present the analytic form of $V_{ijkl}^{\mu\nu}$ in the next subsection.

### 3.1.1 Analytic form of the interaction coefficients

First, we realize that the coefficients for a generic two-body interaction are given by (here and in what follows we omit the spin index $(\mu, \nu)$ without loss of generality)

$$V_{ijkl} = \int dx_1 dx_2 \; \phi_i(x_1) \phi_j(x_2) V(x_1 - x_2) \phi_l(x_1) \phi_k(x_2). \tag{11}$$

In the examples, we will focus on the case $V = g\delta(x_1 - x_2)$, i.e., on the contact interaction between $\uparrow$- and $\downarrow$-particles. However, here, we discuss a general interaction. Notice that the relative and center-of-mass coordinates, $x = (x_1 - x_2)/\sqrt{2}$ and $X = (x_1 + x_2)/\sqrt{2}$, can be decoupled for two interacting particles in a harmonic trap. This decoupling leads to two separate problems: one for the center-of-mass motion, another for the relative motion, which simplifies the problem as $V$ depends only on $x$. In general, the transformation reads as follows

$$\phi_i(x_1)\phi_j(x_2) \equiv \langle x_1 x_2 | \phi_i \phi_j \rangle_C \tag{12}$$

$$= \langle x_1 x_2 | \left[ \sum_{ab} |\phi_a \phi_b\rangle_R \; {}_R\langle \phi_a \phi_b | \right] |\phi_i \phi_j\rangle_C \equiv \sum_{ab} \alpha_{ij,ab} \; \phi_a(x)\phi_b(X) \tag{13}$$

where $|\phi_i\phi_j\rangle_C$ denote two-particle states in the "Cartesian coordinate system" where $i$ and $j$ refer to the quantum numbers in the single-particle basis. $|\phi_a\phi_b\rangle_R$ are the states in the "relative basis" where the indicies $a$ and $b$ fix the quantum numbers for the relative and the center-of-mass motions, respectively. The coefficients $\alpha_{ij,ab} \equiv {}_R\langle \phi_a\phi_b|\phi_i\phi_j\rangle_C$ determine the transformation between these two bases (cf. the Talmi-Moshinsky-Smirnov brackets [27]). Their explicit form is given in App. B.

In the new coordinates the element $V_{ijkl}$ can be written as

$$V_{ijkl} = \sum_{a,a',b} \alpha_{ij,ab}\alpha_{kl,a'b} \; v_{aa'}, \qquad \text{where} \qquad v_{aa'} = \int dx \; \phi_a(x)\phi_{a'}(x)V(\sqrt{2}x). \tag{14}$$

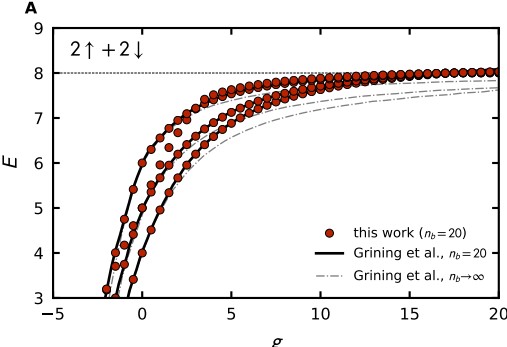 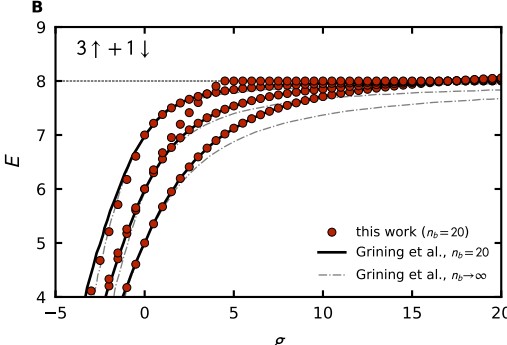

Figure 2: **Benchmark for plain cutoff (Example I).** Panel (A) presents the low-lying energy spectrum for $2 \uparrow + 2 \downarrow$. Panel (B) is for $3 \uparrow + 1 \downarrow$. In both panels, red disks show the lowest part of the energy spectrum obtained in this work for $n_b = 20$, which agrees perfectly with the data from Ref. [26] (thick black curves). For completeness, we also show extrapolated values for $n_b \to \infty$ (dashed-dotted gray curves). The horizontal lines show the expected result at infinite repulsion, $g \to \infty$. The lowest states in panels (A) and (B) should approach this limit as $8 + K_i/g$, where $K_i$ depends on the state [29, 30].

Note that here $a = i + j - b$ and $a' = k + l - b$, since the transformation in Eq. (12) conserves the energy of the state. Consequently, there is only a single sum in Eq. (14):

$$V_{ijkl} = \sum_b \alpha_{ij,(i+j-b)b} \, \alpha_{kl,(k+l-b)b} \, v_{(i+j-b)(k+l-b)}. \tag{15}$$

The separation in the center-of-mass and relative motions is a convenient way to write the coefficients for a general potential, e.g., the effective two-body interaction that we employ in the second part of the present work. For simplicity, we do not compute these coefficients on the fly but pre-compute them with a symbolic language (Mathematica [28]) and read them from a file at the beginning of the program. Note, however, that the $\alpha$ coefficients may also be computed in Julia, if desired. For a $\delta$-potential, the matrix $v_{aa'}$ can be easily calculated: $v_{aa'} = g\phi_a(0)\phi_{a'}(0)/\sqrt{2}$. Note that it is zero if either $a$ or $a'$ is odd, which further simplifies the expression[4].

### 3.1.2 Numerical results

Here, we present some benchmark calculations for two different few-body systems at various coupling strengths. Motivated by the availability of numerically exact data [26], we focus on the four-body systems: $3 \uparrow + 1 \downarrow$ and $2 \uparrow + 2 \downarrow$ at a moderate cutoff of $n_b = 20$. The corresponding size of the many-body basis is presented in Fig. 3. Results for the lowest part of the spectrum for both systems are shown in Fig. 2, where we observe a perfect agreement with the data available in the literature. The energy levels that appear in our data but not in Ref. [26] are likely the center-of-mass excitations that are decoupled from the relative motion in harmonically trapped systems.

The purpose of this section is to highlight a simple use-case for the presented framework. Therefore, we refrain from presenting extrapolations to the infinite basis limit, i.e., $n_b \to \infty$.

---

[4]Note that for a $\delta$-potential the elements $V_{ijkl}$ can be written in other convenient forms, see, e.g., Ref. [5].

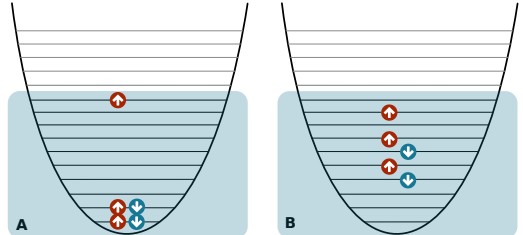

| | $n_b$ | plain cutoff | energy cutoff |
|---|---|---|---|
| $3\uparrow + 1\downarrow$ | 20 | 22800 | 1326 |
| $3\uparrow + 1\downarrow$ | 30 | 121800 | 6360 |
| $3\uparrow + 1\downarrow$ | 40 | 395200 | 19530 |
| $2\uparrow + 2\downarrow$ | 20 | 36100 | 2200 |
| $2\uparrow + 2\downarrow$ | 30 | 189225 | 10200 |
| $2\uparrow + 2\downarrow$ | 40 | 608400 | 30800 |

Figure 3: **Construction of an 'improved' many-body basis.** (Left) Sketch of the states that are considered (A) and discarded (B) due to the energy restriction (see the text for more detail). (Right) Dimension of the many-body basis with and without the energy restriction for various system parameters.

For completeness, we only present the corresponding results from Ref. [26], which show that the plain cutoff with $n_b = 20$ does not lead to accurate results in particular for strong couplings, $g \to \infty$. This motivates a few improvements to the used many-body basis that we discuss in the subsequent examples.

## 3.2 Example II: 1D harmonically trapped fermions with 'energy cutoff'

As a simple extension to the previous example, we consider a more-physically-motivated approach to the choice of the truncation procedure that dramatically reduces the dimensionality of the many-body basis. The main idea is to neglect contributions from Fock states whose non-interacting energy is above a certain threshold $E_{\max}$ (see, e.g., [5,6]). This strategy may be implemented in several ways, however, to facilitate a straightforward comparison with the previous results we define the maximum energy cutoff via the maximally considered SPO[5]. Specifically, we take the maximal energy to be the sum of all single-particle energies in a Fock state obtained by placing all but one particle in the energetically lowest orbitals while the remaining particle is placed in the maximally considered one [5] (this is sketched in Fig. 3). For the case of 1D harmonically trapped fermions, the maximal energy is then given by

$$E_{\max} = \sum_{i=1}^{N_\uparrow - 1} \left(i - \frac{1}{2}\right) + \sum_{i=1}^{N_\downarrow} \left(i - \frac{1}{2}\right) + n_b - \frac{1}{2} = \frac{(N_\uparrow - 1)^2}{2} + \frac{N_\downarrow^2}{2} + n_b - \frac{1}{2}, \quad (16)$$

with the convention $N_\uparrow \geq N_\downarrow$. A comparison of the dimensionality of the many-body basis is shown for a selection of system parameters in the table of Fig. 3. Naturally, the results for a given basis cutoff $n_b$ for the energy cutoff scheme are expected to be worse than those for the plain cutoff, as one merely discards some (ideally unimportant) states. However, the hope is to obtain better results with the energy restriction when going to larger $n_b$ which can either only be reached by massive numerical effort or are out of reach altogether with the plain cutoff.

Within the `FermiFCI.jl` framework the energy restriction may be implemented straightfowardly: One merely needs to provide an alternative list of Fock states to the solver, everything else will be handled automatically. Therefore, the only change in an actual implementation of this example is an additional function that finds the energy-restricted subspace of

---

[5]This convention also circumvents the subtlety of requiring large indicies in the single-particle basis when one employs the inverse approach, i.e., when a maximal energy cutoff is fixed and then the index of the maximally required SPO is computed. While this is not a problem in principle, there could be issues with precision of data types.

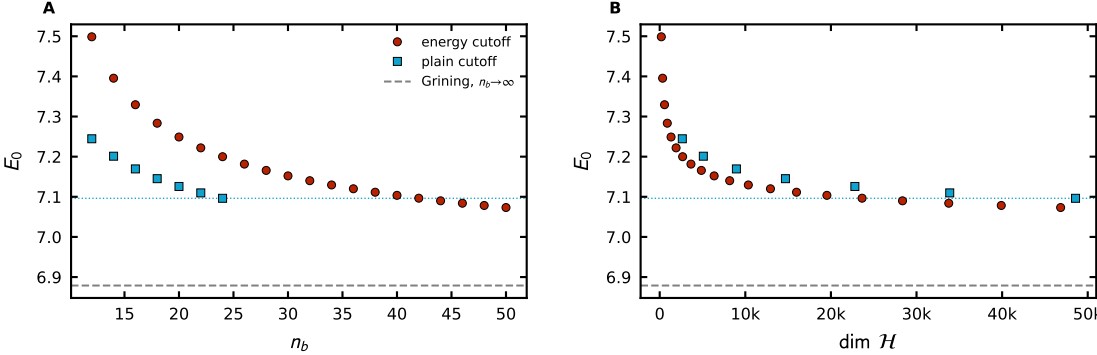

Figure 4: **The plain cutoff vs. the energy cutoff for $3\uparrow+1\downarrow$ with $g=5.0$: Energies** (A) The ground-state energy as a function of the basis cutoff for the energy-restricted Hilbert space (red disks) and a plain cutoff (blue squares). (B) The ground-state energy as a function of the dimensionality of the many-body basis. The dashed horizontal lines show the result in the limit $n_b\to\infty$. The dotted horizontal lines is a guide to the eye to facilitate a comparison to the lowest energy obtained using plain cutoff (at $n_b=24$).

the Hilbert space obtained with plain cutoff. Finally, it is worth pointing out that an ED with any other restriction on the Hilbert space follows from the procedure outlined in this section. Therefore, minor changes in the actual FCI implementation allow one to quickly test and benchmark novel approches with little to no overhead.

### 3.2.1 Benchmark: energy cutoff vs. plain cutoff

To illustrate that the energy cutoff scheme is superior to the plain cutoff, we present numerical results for $3\uparrow+1\downarrow$ with $g=5.0$ in Fig. 4. In panel (A) we show the convergence of the ground-state energy as a function of $n_b$, where we can immediately see that for a given $n_b$ the energy cutoff leads to the values that are slighlty worse than those for the plain cutoff, as expected. However, the energy cutoff leads to more accurate results for the same dimensionality of the Hilbert space. This is highlighted in panel (B) of Fig. 4, where the ground-state energy is shown as a function of $\dim\mathcal{H}$: Clearly, for a fixed value of $\dim\mathcal{H}$, the energy cutoff leads to the energies that are below those of the plain cutoff. Nevertheless, both approaches are still some way off the extrapolated results from Ref. [26], which require even larger Hilbert spaces. While drastically larger values of $n_b$ are certainly prohibitive for the plain cutoff, the ED with the energy cutoff may be pushed further and be more suitable for an accurate extrapolation to $n_b\to\infty$.

### 3.2.2 Density profiles

Here, we illustrate a calculation of observables beyond the energy and their convergence to $n_b\to\infty$. To this end, we analyze density profiles. These can be obtained from the one-body density matrix, defined as

$$\rho_{ij}^{\sigma}=\langle\psi_0|\hat{\psi}_{\sigma,i}^{\dagger}\hat{\psi}_{\sigma,j}|\psi_0\rangle=\sum_{n,m}c_n^*c_m\,\langle\phi_n|\hat{\psi}_{\sigma,i}^{\dagger}\hat{\psi}_{\sigma,j}|\phi_m\rangle \tag{17}$$

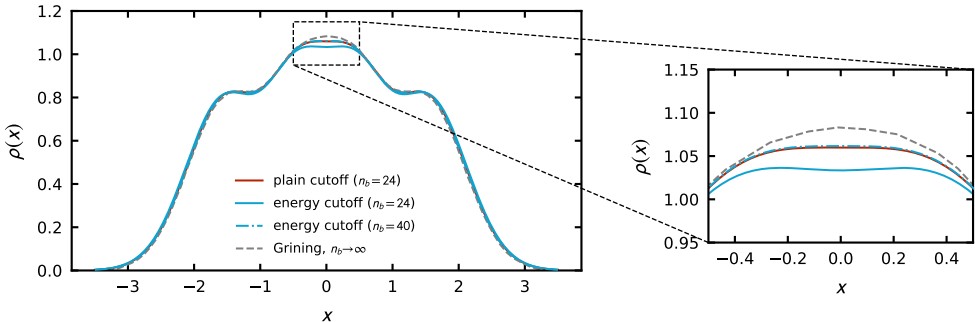

Figure 5: **The plain cutoff vs. the energy cutoff for** $3\uparrow+1\downarrow$ **with** $g=5.0$**: Densities** The main graph shows the total density profiles ($\rho_\uparrow+\rho_\downarrow$) obtained with plain cutoff at $n_b = 24$ (red solid) together with those based upon the energy cutoff at $n_b = 24$ (blue solid) and $n_b = 40$ (blue dot-dashed). We also present results from Ref. [26] (dashed gray curve). The inset shows a zoomed region to the center of the trap, where the differences are the largest.

where $|\psi_0\rangle = \sum_n c_n|\phi_n\rangle$ denotes the (normalized) ground-state wavefunction, and $\sigma$ is either $\uparrow$ or $\downarrow$. The density for a given spin component is then simply given by

$$\rho_\sigma(x) = \sum_{ij} \rho_{ij}^\sigma \, \phi_i^*(x)\phi_j(x) \tag{18}$$

where $\phi_i(x)$ is the spatial representation of the $i$-th single-particle orbital, see Eq. (10). Our package provides a tool for computing both of these quantities, such that a computation might look like:

```
1    # First, compute the one-body density matrix for the up component (flavor=1).
2    flavor = 1
3    obdm_up = FermiFCI.compute_obdm(wf, flavor, lookup, inv_lookup, n_basis)
4
5    # Now we fix the spatial grid and compute the spatial density profile.
6    x_grid = collect(-3.5:0.01:3.5)
7    rho_up = FermiFCI.compute_density_profile(HOOrbital1D, x_grid, obdm_up)
```

Results for the spatial density profiles are benchmarked against the values from Ref. [26] in Fig. 5. Again, we observe that for $n_b = 24$ the results of the ED method with the energy cutoff are slightly worse than those of the ED with the plain cutoff. However, for $n_b = 40$ the results of the ED method with the energy cutoff are virtually undistinguishable from the plain-cutoff result albeit at a reduced numerical effort (cf. Fig. 4). Figure 5 demonstrates that in the center of the trap both approaches still show some discrepancy to the result from Ref. [26]. The difference can be resolved by employing larger cutoffs or by introducing further improvements (see, e.g., our discussion of the effective interaction in Sect. 4).

### 3.3   Example III: Particles in a flat trap

For our third illustration of the `FermiFCI.jl` framework, we consider a few fermions in a flat trap (i.e., hard-wall trap) where the species have different masses, i.e., a mixture of different

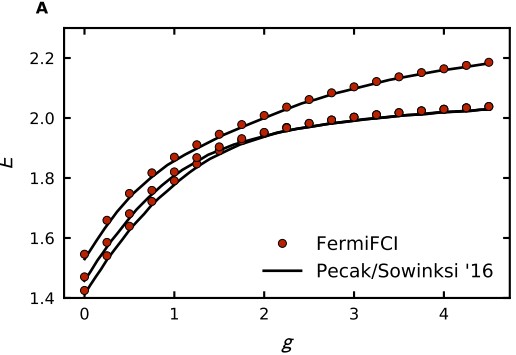
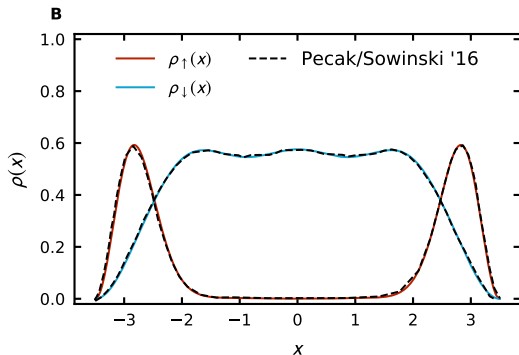

Figure 6: **The $1\uparrow+3\downarrow$ system with mass imbalance in a box (Example III).** (A) The lowest part of the spectrum. Red discs are calculated in this work. Black curves represent the values from Ref. [31]. We use the plain cutoff with $n_b = 14$. (B) The density profiles for the up (red) and down (blue) components in a flat box of extent $[-3.5, 3.5]$ compared to those of Ref. [31] at $g = 4.0$.

atomic species. The corresponding Hamiltonian reads

$$\hat{H} = \sum_{\sigma=\uparrow,\downarrow} \sum_{i=1}^{N_\sigma} \left[ -\frac{\hbar^2}{2m_\sigma}\nabla^2_{x_i} + V_{\text{flat}}(x_i) \right] + \frac{g}{2}\sum_{i\neq j} \delta(x_i - x_j) \tag{19}$$

with the hard-wall potential $V_{\text{flat}}(x) = 0$ for $|x| < L$, otherwise $V_{\text{flat}}(x) \to \infty$. The particle-particle interactions are again of the $s$-wave type and occur only between $\uparrow$- and $\downarrow$-particles.

The main difference of the implementation to the above examples is the nature of the single-particle orbitals, which we now take to be eigenstates of the one-body problem in a flat trap. The spatial representation for the orbital with the energy

$$E_n^\sigma = \frac{\hbar^2 \pi^2 n^2}{8m_\sigma L^2} \tag{20}$$

reads

$$\phi_n(x) = \frac{1}{\sqrt{L}} \sin\left[\frac{n\pi}{2L}(x+L)\right], \tag{21}$$

where the convention is such that the lowest orbital corresponds to $n = 1$ and $m_\sigma$ denotes the mass of a particle of spin $\sigma$. In the following, we use $m_\downarrow = 1$ as our reference scale, in accordance to Ref. [31] which we use for benchmarking. To use these states in the input of our framework we define a new datatype that reflects the expressions from above ($\hbar = 1$):

```
1   abstract type BoxOrbital <: Orbital end
2
3   struct BoxOrbital1D <: BoxOrbital
4       L :: AbstractFloat # Half the size of the box, domain is [-L,L].
5       m :: AbstractFloat # Mass of the particle.
6   end
7
8   # Callable for the energy.
9   (orb::BoxOrbital1D)(n::Integer)::AbstractFloat = begin
10      return n^2*pi^2/8/(orb.L^2)/orb.m
11  end
```

```
12
13     # Callable for the spatial representation of the wavefunction.
14     (orb::BoxOrbital1D)(n::Integer, x::Number)::AbstractFloat = begin
15         return 1.0/sqrt(orb.L)*sin(pi*n*(x+orb.L)/(2*orb.L))
16     end
```

Aside from the definition of the data type, one needs to fix the one-body ($C_{ij}$) and two-body ($V_{ijkl}$) coefficients which can be done in a very similar way as for the example presented in Sect. 3.1.

In Fig. 6 we show a benchmark for a $1+3$ system at a cutoff of $n_b = 14$ and $m_\uparrow/m_\downarrow = 40/6$, in accordance with the values presented in Ref. [31]. To be consistent with the available data, we set the length scale (i.e., half the length of the box) to $L = 3.5$. As in the previous examples, the lowest part of the spectrum agrees perfectly with the benchmark data. Finally, we also show the density profiles for the spin-up and -down particles in panel (B) of Fig. 6. For strong interactions, the spin-up particles separate from spin-down particles in accord with Ref. [31]. The physics of this separation is understood by noticing that the interaction energy can be minimized if the two species separate. It is easier to push the heavy impurity to the edge, since the kinetic energy of the heavy impurity is small (see also Ref. [32]).

## 4    Effective two-body interaction

This section is devoted to an effective two-body interaction. Namely, we discuss a physically motivated choise of the elements $V_{ijkl}$ that may be straightforwardly accomodated in the generic `FermiFCI.jl` framework. We first give an intuitive overview of the underlying ideas, which apply to any system. Then, we provide an explicit construction of the effective interaction for 1D harmonically trapped fermions and show that an FCI calculation with this interaction yields accurate results at considerably reduced computational effort.

### 4.1    Physical motivation

The procedure employed in the preceding sections, namely the gradual increase of the many-body basis size at a fixed coupling parameter $g$, allows one to extract well-converged results for one-dimensional systems, as was exemplified for the harmonic oscillator with a $\delta$-interaction. However, an accurate extrapolation to the infinite basis result is out of reach for many cases of interest[6]. Luckily, the poor situation with convergence can often be improved by proper renormalization of the coupling. To this end, one has to find a relation between the physics in the truncated (regularized) Hilbert space and the "real" physics that formally takes place in the infinite basis. While the former is dictated by the (bare) coupling strength $g$, the latter is governed by the $s$-wave scattering length $a_0$. The mapping between these two quantities is non-trivial and in general depends on the type of the applied truncation. For cases where an exact solution for the two-body problem is available, such as for particles in a harmonic trap [16], a convenient procedure is to compute the two-body energy at a fixed cutoff in the restricted model space and then match it to the exact expression which in turn maps the

---

[6]In fact, in dimensions $d > 1$ such a simple approach might fail to capture the correct physics altogether. In particular, the $\delta$-interaction requires a careful treatment in spatial dimensions higher than one, see, e.g., Refs. [6, 13, 33–35].

energy to the scattering length[7]. In this way, a map between the physics of the effective model and the "real" system is established. Such approach has proven to be useful and was recenlty explored for harmonically trapped fermions in 2D [6, 34].

Jimmy Rotureau in Ref. [13] puts forward a similar regularization method for a harmonically trapped system in 3D, which results in an effective two-body interaction in the restricted model space. The key idea is to use the exact two-body solution and the corresponding analytically known eigenfunctions [16]. By construction, the resulting effective two-body interaction exactly reproduces the continuum values of the two-body problem already in the restricted space. So, instead of using only the lowest energy value, an extended region of the energy spectrum is employed to renormalize the coefficients $V_{ijkl}$. Although technically this constitutes an uncontrolled approximation for larger sytems, it is a systematically improvable one. Remarkably, the convergence properties with the basis cutoff are vastly improved (see Example IV).

## 4.2    Construction of the effective interaction matrix

Here, we sketch the construction of such an effective interaction mainly following Ref. [13]. To this end, we consider an arbitrary two-body problem whose matrix elements we write in an arbitrary Fock basis as follows

$$H_{ab}^{(2)} = \langle \Phi_a | \hat{H}^{(2)} | \Phi_b \rangle, \tag{22}$$

where $|\Phi_a\rangle$ denotes a (properly symmetrized) many-body state constructed with the appropriate single-particle orbitals. The superscript '(2)' indicates a two-body problem.

We assume that this two-body problem is analytically (or numerically) solvable, and we use its eigenfunctions $\hat{H}^{(2)} |\Psi_n\rangle = \varepsilon_n |\Psi_n\rangle$ to rewrite the matrix elements from Eq. (22) as follows

$$H_{ab}^{(2)} = \sum_{n,m} \langle \Phi_a | \Psi_n \rangle \langle \Psi_n | \hat{H}^{(2)} | \Psi_m \rangle \langle \Psi_m | \Phi_b \rangle \tag{23}$$

$$= \sum_n \varepsilon_n \langle \Phi_a | \Psi_n \rangle \langle \Psi_n | \Phi_b \rangle. \tag{24}$$

This expression may conveniently be re-written as the unitary transformation

$$H_{ab}^{(2)} = \sum_{n,m} (U^\dagger)_{an} E_{nm}^{(2)} U_{mb}, \tag{25}$$

where the matrix $E_{nm}^{(2)} = \delta_{nm} \varepsilon_m$ contains the exact eigenenergies on the diagonal. The basis transformation between the eigenbasis of $\hat{H}^{(2)}$ and an arbitrary basis is parameterized by the matrix $U_{na} = \langle \Psi_n | \Phi_a \rangle$.

Now, we divide the Hilbert space into two parts: the model space $\mathcal{M}$ and the discarded space $\mathcal{D}$. For example, one can restrict $\mathcal{M}$ to the $n_{max}$ energetically lowest eigenstates of $\hat{H}^{(2)}$. This step resembles the single-particle cutoff $n_b$ in the discussion above, see, e.g., Sects. 3.1 and 3.3. While this partition may in principle be done arbitrarly, the presented version is the most straighforward and instructive procedure.

---

[7]For systems in a homogeneous periodic box the appropriate expression would be the so-called Lüscher formula that relates the scattering length to the low-lying spectrum [36, 37].

With this partition we may introduce an effective interaction, which is motivated by the approaches commonly used in nuclear physics, see, e.g., [38–40]. To this end, we define the matrix (the row-column indicies are omitted here and in what follows)

$$Q_{\mathcal{M}} = \frac{U_{\mathcal{M}\mathcal{M}}}{\sqrt{U_{\mathcal{M}\mathcal{M}}^{\dagger} U_{\mathcal{M}\mathcal{M}}}} \tag{26}$$

where the reduced coefficient matrix $U_{\mathcal{M}\mathcal{M}}$ is obtained by projection to the model space $U_{\mathcal{M}\mathcal{M}} = P_{\mathcal{M}} U P_{\mathcal{M}}$ with the projection operator $P_{\mathcal{M}}$. Consequently, we define

$$H_{\mathcal{M}}^{\text{eff}} \equiv Q_{\mathcal{M}}^{\dagger} E_{\mathcal{M}\mathcal{M}}^{(2)} Q_{\mathcal{M}} \tag{27}$$

which converges to the true two-body Hamiltonian once the subspace $\mathcal{M}$ approaches the full Hilbert space, i.e., in the limit $\mathcal{M} \to \mathcal{H}$. In this limit, which reflects full basis, the matrix $U_{\mathcal{M}\mathcal{M}}^{\dagger} U_{\mathcal{M}\mathcal{M}}$ converges to $\mathbb{1}$, leaving us again with the conventional unitary transformation of Eq. (25). At finite cutoff, Eq. (27) may be viewed as the properly normalized unitary transformation in the restricted model space.

To finally obtain an effective two-body interaction in the restricted model space, we need to subtract the kinetic two-body energy in the many-body basis of choice, such that

$$V_{\mathcal{M}}^{\text{eff}} = Q_{\mathcal{M}}^{\dagger} E_{\mathcal{M}\mathcal{M}}^{(2)} Q_{\mathcal{M}} - P_{\mathcal{M}} H_0 P_{\mathcal{M}} = H_{\mathcal{M}}^{\text{eff}} - T_{\mathcal{M}}^0. \tag{28}$$

where $H_0$ is the kinetic (i.e., bilinear or non-interacting) term in our two-body Hamiltonian such that $T_{\mathcal{M}}^0$ is simply a diagonal matrix with the corresponding non-interacting energies on the diagonal.

The above matrix essentially provides an optimized interaction between orbitals in the *two*-particle basis, that is, in the basis of direct products of single-particle orbitals. In other words, the constructed matrix reproduces the lowest part of the exact two-body spectrum in a finite model space. Finally, note that this procedure is not limited to a specific type of the two-body problem. For example, the procedure has been used to analyze fermionic and bosonic systems in a harmonic trap [17, 41] as well as bosonic systems in a hard-wall potential [42]. Notice that the effective interaction is expected to lead to the most noticeable speed-up of the ED method for a harmonic trap, where the convergence of the energy as a function of $n_b$ is slow for $\delta$-interactions. It is proportional to $1/\sqrt{n_b}$ in the limit $n_b \to \infty$ [26].

## 4.3 Example IV: Effective interaction for the 1D harmonic oscillator

So far, our discussion of the effective interaction approach has been agnostic to the specific problem at hand as well as to the spatial dimensionality. In the following we present an explicit construction of the effective two-body interaction for a harmonically trapped system in 1D and perform an extensive benchmark against the existing data. The corresponding implementation of this example within the `FermiFCI.jl` framework is included in the freely available source code [19]. Note that the possibility to separate the center-of-mass motion from the relative motion implies that the effective interaction for particles in the harmonic trap can be constructed either at the level of $V_{ijkl}$ or $v_{aa'}$ (cf. Eq. (14)). Here, we will focus on $v_{aa'}$, since it allows us to illustrate the main steps in a more concise manner.

Note that the use of the effective interaction is not at all necessary for 1D systems, where the exact relation between the scattering length and the coupling is known and one may

extrapolate results to infinite cutoff $n_b$ without worrying (too much) about renormalization. However, as we shall see below, also for fermions in 1D setups the convergence properties are greatly improved which makes few-body computations much more feasible.

### 4.3.1 Explicit construction for the 1D harmonic oscillator

As discussed in 3.1.1, the harmonic oscillator problem is separable in the center-of-mass and the relative parts. Therefore, we may focus on the effective interaction between two orbitals in the relative coordinates. The corresponding effective interaction is constructed by exchanging $v_{aa'}$ in Eq. (14) by the effective interaction (cf. Eq. (28)). All other parts in the FCI implementation stay exactly as in the case of the bare interaction.

To construct the effective interaction, we notice that the relative wavefunction for two particles in a hamornic oscillator is given by [16]

$$f_\nu(x) = \mathcal{N}_\nu e^{-x^2/2} \mathcal{U}(-\nu, \tfrac{1}{2}, x^2) \tag{29}$$

with $\mathcal{U}$ denoting the Tricomi function, $\nu = \frac{\varepsilon}{2} - \frac{1}{4}$, and $x = (x_1 - x_2)/2$. The normalization coefficient reads

$$\mathcal{N}_\nu = \left[ \int dx \; e^{-x^2} \mathcal{U}(-\nu, \tfrac{1}{2}, x^2)^2 \right]^{-1} = \sqrt{\frac{1}{\pi} \frac{\Gamma(-\nu + \tfrac{1}{2})\Gamma(-\nu)}{F(-\nu + \tfrac{1}{2}) - F(-\nu)}} \tag{30}$$

with the digamma function $F(x) = \Gamma'(x)/\Gamma(x)$. The eigenenergies $\varepsilon_n$ are given by the solutions of the equation [16]

$$\frac{\Gamma(-\nu_n + \tfrac{1}{2})}{\Gamma(-\nu_n)} = -\frac{g}{2\sqrt{2}}. \tag{31}$$

Note that analytic solutions for two harmonically trapped fermions also exist in dimensions $d > 1$ (see, e.g., [16]). Therefore, one can in principle address two- and three-dimensional systems using the same routine that we outline here for 1D. However, one might need to adapt in calculations the cutoff strategy.

To construct the effective two-body interaction, we need to explicitly construct the transformation matrix $U^{1D}$. After separating the problem into center-of-mass and relative parts, the problem reduces to computing the overlap between states of the single-particle HO basis in Eq. (10) and the radial solution of the interacting two-body problem given in Eq. (29). Note that odd-parity wavefunctions exhibit a node at the origin and therefore only even-parity states are modified by the interaction. Consequently, we may write the transformation matrix as

$$U_{ij}^{1D} = \begin{cases} \int_{-\infty}^{\infty} dx \; f_{\nu_i}(x)\phi_j(x) & \text{for } i \text{ even and } j \text{ even }, \\ \delta_{ij} & \text{for } i \text{ odd and } j \text{ odd }, \\ 0 & \text{otherwise.} \end{cases} \tag{32}$$

The matrix indicies denote the $i$-th element in the bases, *including* the odd-parity waves, i.e., only even values correspond to the $n$-th solution of the Busch formula in Eq. (31). The integral $\int_{-\infty}^{\infty} dx \; f_{\nu_i}(x)\phi_j(x)$ can be explicitly calculated by exploiting the fact that $f_\nu$ is an eigenfunction of the Schrödinger equation, thus,

$$U_{ij}^{1D} = -g \frac{f_{\nu_i}(0)\phi_j(0)}{E_j - \epsilon_{\nu_i}} \quad \text{for } i \text{ and } j \text{ even,} \tag{33}$$

where $E_j = j + \frac{1}{2}$ denotes the total energy of the $j$-th single particle orbital $\phi_j$, which is introduced in Eq. (10). Note that for the 1D harmonic oscillator the matrix of the single-particle Hamiltonian simply reads $T_{ij}^0 = \delta_{ij}\,(i + \frac{1}{2})$, which completes the necessary ingredients for the construction of $V^{\text{eff}}$ according to Eq. (28).

In the following we highlight the usefulness of the effective interaction by benchmarking to the available numerical [26,43] and experimental results [23,44]. An overview of our benchmark is shown in Fig. 7, where we present results for the convergence of a $3 \uparrow +1 \downarrow$ system (chosen mainly because of the already existing numerically exact data – similar conclusions hold for other particle numbers). Note that all of the calculations shown in Fig. 7 amounted to less than 30 minutes of computation time on a standard notebook *in total*.

### 4.3.2    Energy spectrum

As a first step we compare the lowest part of the spectrum for couplings $g \in [0, 25]$, which correspond to the transition from the non-interacting case to the strongly repulsive regime. We show the interaction energy $E_{\text{int}} \equiv E - E_{\text{F}}$, with the total energy $E$ and $E_{\text{F}} = \frac{1}{2}\sum_\sigma N_\sigma^2$, for a small basis cutoff $n_b = 14$ in panel A of Fig. 7, for both the bare and effective interactions. The advantage of the effective interaction is immediately apparent, as the results essentially agree with the numerically exact results of [26]. For the intermediate coupling, $g = 5.0$, the convergence with the basis cutoff is shown in the inset. While the bare interaction leads to relative uncertainties w.r.t. $E_{\text{int}}(n_b \to \infty)$ between $10\% - 20\%$, the effective interaction leads to relative uncertainties $\lesssim 1\%$ for *all* cutoffs shown in the plot.

**Benchmark against extrapolated values and experiment**    In addition to the benchmark against the state-of-the-art numerical few-body results, we compare our findings to the experimental data for few-fermion systems in 1D microtraps [23,44]. In panel C of Fig. 7, we illustrate this comparison for systems with $(N-1) \uparrow +1 \downarrow$ particles, which provide information about the phase diagram of highly imbalanced mixtures of atoms, see, e.g., Ref. [45,46] and references therein. Our calculations are performed with a very small basis-cutoff of $n_b = 10$. Remarkably, we find excellent agreement with the experimentally studied (relatively weak) coupling strenghts for particle numbers up to $N = 6$, which have also been found to agree with earlier FCI results [26]. Due to a fast convergence of our results we are able to easily extend this range to $N = 9$ while still being far away from limitations of computational resources.

As a second benchmark we consider the separation energy

$$\Delta E = \mu(N) - \mu_0(N), \tag{34}$$

where $\mu(N) = E(N) - E(N - 1)$ is the chemical potential; $\mu_0(N)$ is its non-interacting value. Again, we report excellent agreement with earlier theoretical determinations of the separation energy [26,47] for attractive systems at $g = -0.4$, as shown in panel D of Fig. 7. The 'even-odd oscillation' visible in the figure suggests a BCS-like pairing, see also Ref. [47]. Although the experimental values [23] have been measured at $g \approx -0.6$, we observe better agreement between the theoretical values and experiment for slightly smaller coupling of $g = -0.4$, which confirms the observations in Refs. [26,47].

Again, due to the excellent convergence properties, with the effective interaction we are able to extend the range up to $4 \uparrow +4 \downarrow$ particles in a 1D harmonic trap on a standard

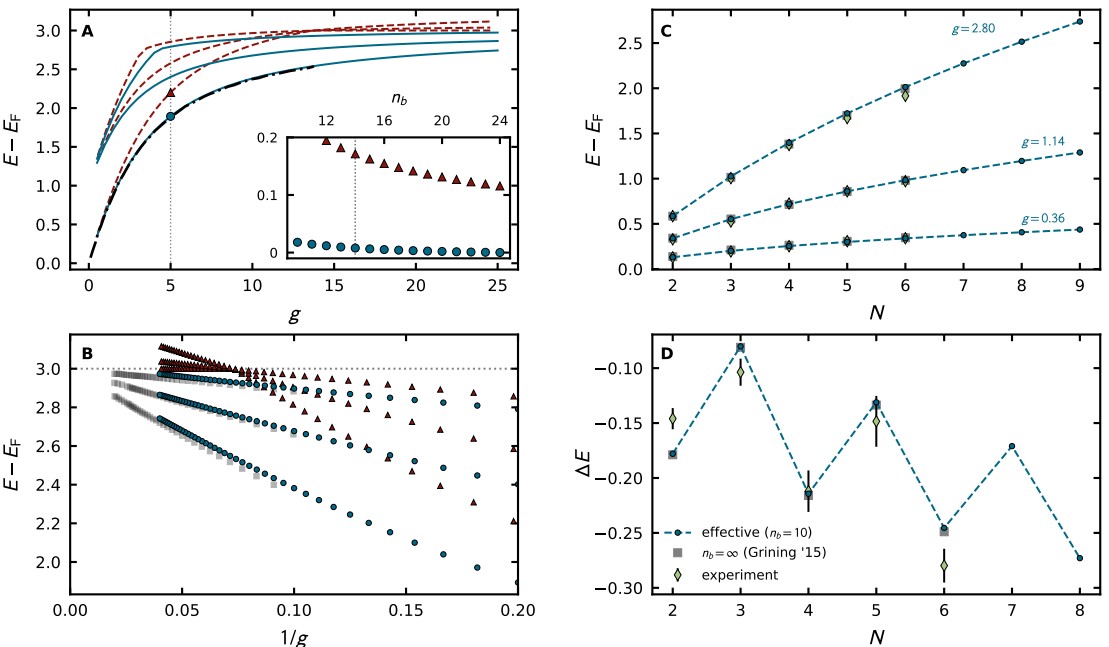

Figure 7: **Summary of the benchmark of the effective interaction against existing data.** The data in the left panels (A,B) correspond to the system $3 \uparrow +1 \downarrow$. Our data are benchmarked against Ref. [26]. (A) The lowest part of the energy spectrum as a function of the interaction strength, for both bare (dashed red curve) and effective interaction (solid blue curve) assuming $n_b = 14$. The numerically exact result for the ground-state energy in the limit $n_b \to \infty$ is shown as a dash-dotted black curve [26]. (Inset A) Relative error of the ground-state energy for $g = 5$ w.r.t to the limit $n_b \to \infty$ as a function of the cutoff parameter $n_b$ for bare (red triangles) and effective (blue disks) interactions. (B) The three lowest energy levels in the strongly repulsive limit $g \to \infty$ for the bare (red triangles) and effective (blue disks) interactions for $n_b = 14$. Gray squares show the extrapolated FCI values (i.e., $n_b \to \infty$) from [26]. (C) The interaction energy for the system $(N-1) \uparrow +1 \downarrow$ obtained with the effective interation (blue disks). We also present the extrapolated FCI values from [26] (gray squares) as well as experimental measurements [44] (green diamonds). (D) The separation energy $\Delta E$ obtained using the effective interaction (blue disks) compared to the extrapolated FCI results from [26] (gray squares) and experiment [23] (green diamonds). Lines in panels (C) and (D) are added to guide the eye.

notebook. A few more particles are possible with reasonable resources when running those calculations on a supercomputer.

### 4.3.3   Density profiles

Besides the ground-state energy, as discussed above, the density profile $\rho(x)$ is a central quantity of interest for trapped Fermi gases. In the following we show that - contrary to a naive guess - this quantity does not converge (everywhere) faster than the ground-state energy and illustrate this statement with numerical data.

To understand the convergence properties of the density, we first consider the convergence of the density in the relative coordinates. In this case, the function from Eq. (29) is conveniently written as

$$f_\nu(x) = gf_\nu(0) \sum_{n=0}^{\infty} \frac{\phi_n(0)\phi_n(x)}{n + \frac{1}{2} - \epsilon_\nu}, \tag{35}$$

where $\epsilon_\nu$ is the energy that corresponds to $f_\nu$. An approximation to this function in a finite basis, $f_\nu^{(n_b)}(x)$, reads as follows

$$f_\nu^{(n_b)}(x) = gf_\nu^{(n_b)}(0) \sum_{n=0}^{n_b} \frac{\phi_n(0)\phi_n(x)}{n + \frac{1}{2} - \epsilon_\nu^{(n_b)}} \simeq gf_\nu^{(n_b)}(0) \sum_{n=0}^{n_b} \frac{\phi_n(0)\phi_n(x)}{n + \frac{1}{2} - \epsilon_\nu} + \frac{\Delta(x)}{\sqrt{n_b}}, \tag{36}$$

where $\Delta(x)$ is some function, and we used the known form of the convergence of the energy $\epsilon_\nu^{(n_b)} = \epsilon_\nu + O(1/\sqrt{n_b})$ [26]. Using the expressions above, we derive

$$\frac{f_\nu(x)}{f_\nu(0)} - \frac{f_\nu^{(n_b)}(x)}{f_\nu^{(n_b)}(0)} \simeq g \sum_{n=n_b}^{\infty} \frac{\phi_n(0)\phi_n(x)}{n + \frac{1}{2} - \epsilon_\nu} - \frac{\Delta(x)}{\sqrt{n_b}f_\nu^{(n_b)}(0)}. \tag{37}$$

Note that only the harmonics with $n = 2k$ contribute to the sum of Eq. (37). To estimate this sum, we consider the auxiliary quantity $S(k) = \phi_{2k}(0)^2/(2k)$, which equals to

$$S(k) = \frac{1}{\sqrt{\pi}} \frac{(2k-1)!}{4^k (k!)^2}. \tag{38}$$

This expression is obtained by using $H_{2k}(0) = (-1)^k (2k)!/k!$ in Eq. (10). To proceed, we use Stirling's approximation, $n! \sim \sqrt{2\pi n}(n/e)^n$, which leads to

$$S(k) \sim \frac{1}{k^{3/2}}. \tag{39}$$

This tail of $S(k)$, implies that the wave function in relative coordinates converge as $1/\sqrt{n_b}$ $[\int_{k_{max}}^{\infty} S(k)dk \sim 1/\sqrt{k_{max}}]$, i.e., as slow as the energy [26], at least in the vicinity of the meeting point of two particles. Therefore, we write

$$f_\nu^{(n_b)}(x) \simeq f_\nu(x) + \frac{\tilde{\Delta}(x)}{\sqrt{n_b}}, \tag{40}$$

where $\tilde{\Delta}(x)$ is some function of finite support. It is straightforward to show that this form of the function in the relative coordinates leads to a slow convergence ($\sim 1/\sqrt{n_b}$) of the two-body density. The physical understanding of the slow convergence can be based on the fact that

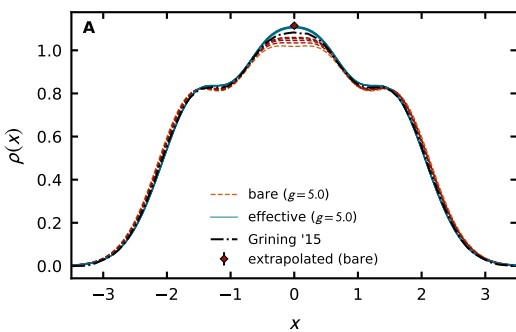 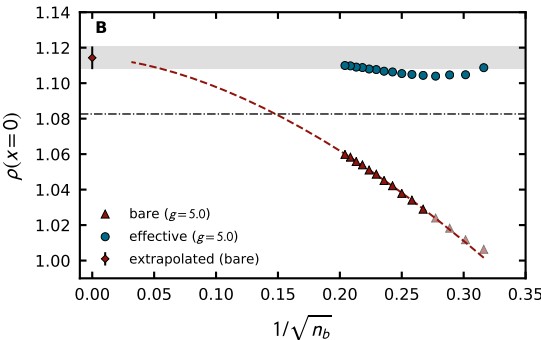

Figure 8: **Calculation of the density using the effective interaction for a** $3\uparrow +1\downarrow$ **system with** $g = 5$. (A) Density profiles obtained with bare (dashed red curves) and effective (solid blue curves) interaction compared to the results of Ref. [26] with a finite cutoff (dash-dotted black curve). (B) Convergence of the density at the center of the trap with $n_b$. The fitted function (dashed curve) converges to the symbol at $1/\sqrt{n_b} = 0$. The gray band shows the uncertainty from the fitting procedure. The horizontal dash-dotted line is the reference value from Ref. [26], which was likely obtained at a finite cutoff $n_b$.

to represent the non-analyticity (cusp) of the wave function, one needs infinitely many states $\phi_n(x)$. A finite cutoff parameter effectively leads to a finite range of the potential of the order of $1/\sqrt{n_b}$, which may modify observables in the same order.

We corroborate this notion with a numerical determination of $\rho(x)$ for a $3\uparrow +1\downarrow$ system at $g = 5$, see panel A of Fig. 8. The dashed red curves correspond to densities obtained with the bare interaction. We observe a relatively slow convergence with increasing single-particle cutoff $n_b$ (curves from the bottom to the top correspond to increasing values of $n_b \in [12, 24]$). In panel B of the same figure, the convergence is explicitly shown for the density at the origin, $\rho(x = 0)$, alongside an extrapolation to $n_b \to \infty$ with the functional form $\rho(n_b) = a + bn_b^c$ (dashed line). It must be noted that the extrapolated value slightly depends on the number of points used in the extrapolation (here $n_b \geq 14$). From our numerical data for the bare interaction we observe that $c \approx -0.7$, reflecting a slightly faster convergence than the one for the ground-state energy (which, to leading order, exhibits $c = -0.5$). This might be an artifact of a small number of the used values of $n_b$ in the fitting procedure. In any case, fully converged results for the density profiles still require calculations at large single-particle cutoffs $n_b$, as evident from our data.

Finally, we show results obtained with the effective interaction (blue curves / symbols). As for the energy, we observe rapid convergence of the density with $n_b$ which on the scale of panel A cannot even be resolved. A closer look is again presented in panel B of Fig. 8, where we observe that the effective interaction is essentialy within the uncertainty of the extrapolated value from the bare interaction (gray shaded band). We note that the result for the density presented in [26] (black dashed-dotted line) differs slightly from our prediction, likely because the reference value was obtained at a finite cutoff.

# 5 Discussion & Outlook

In conclusion, we provide a generic and easy-to-use ED framework in the context of strongly-interacting trapped few-fermion systems. This framework should serve as a testbed for newcomers to the field and lower the barrier for exploration of new optimization and approximation schemes. To illustrate the versatility of our code, we explore a handful of examples, which highlight the fact that a simple tool for testing new ideas quickly can lead to substatial improved results and reduced computational cost. Moreover, we provide an in-depth discussion of an effective two-body interaction approach that we also implement in the presented framework. A detailed benchmark for systems and observables previously not treated with such a method revealed a dramatic improvement of the convergence properties and allowed us to deliver state-of-the-art numerical values at significantly lower cost than with the commonly used bare interaction.

For the sake of clarity, all examples considered in this work are in one spatial dimension. However, there is no intrinsic reason that the generic implementation cannot be used for higher-dimensional systems. A prime example, for instance, would be the application of our framework to experimentally accessible few-body systems in 2D [48, 49], which, to date, present a challenge for numerical analysis beyond $3\uparrow +3\downarrow$. Note that the effective two-body interaction was initially proposed for three-dimensional systems [13], however, there should be no obstacles to adopt the method for calculations in 2D. In addition to the standard fermionic use-case our generic application allows – in principle – the application to bosonic few-body problems through the inclusion of intra-species couplings. It must be noted, however, that such a calculation would be quite limited in basis size and hence would mainly serve as benchmark for small systems.

Apart from the immediate application in deterministic FCI calculations, the effective interaction approach might also be useful in other numerical methods. One example is the stochastic calculations for trapped fermions. Although the optimized matrix might not be sign-preserving, methods exist to potentially mitigate or circumvent this issue such as the stochastic FCI approach [10, 50] or the complex Langevin method [51] which so far have mostly been used for systems with periodic boundary conditions. Future studies might also investigate extensions of the effective-interaction approach in the context of the state-of-the-art calculations of time dynamics [52, 53] whose convergence properties depend strongly on particle-particle interactions, see, e.g., [54].

# Acknowledgements

We acknowledge fruitful discussions with Hans-Werner Hammer and thank Gerhard Zürn and Pietro Massignan for sending us their data. We thank Fabian Brauneis for beta-testing the provided code-package, and comments on the manuscript.

**Funding information** L.R. is supported by FP7/ERC Consolidator Grant QSIMCORR, No. 771891, and the Deutsche Forschungsgemeinschaft (DFG, German Research Foundation) under Germany's Excellence Strategy –EXC–2111–390814868. A.G.V. acknowledges support by European Union's Horizon 2020 research and innovation programme under the Marie Skłodowska-Curie Grant Agreement No. 754411.

# A  Technical details for the Hamiltonian construction

In this appendix we briefly elaborate on the construction of the Hamiltonian matrix as implemented in the `FermiFCI.jl` package. As a prerequisite, we assume that the appropriate Hilbert space (i.e., the many-body basis discussed in Sect. 2.3.1) has been assembled and the corresponding one- and two-body matrix-elements for the problem at hand are pre-computed.

The construction of the Hamiltonian matrix is essentially achieved through looping over all states in the Hilbert space, which we denote by the set $\{|\psi_n\rangle\}$. To each state $|\psi_i\rangle$ we then apply the full Hamiltonian such that a set of new states $\{|\psi_j\rangle\}$ is created, one for each non-zero term of the Hamiltonian. For each of these new states we collect the indicies $i$ and $j$ as well as the matrix element $h_{ij} = \langle\psi_i|\hat{H}|\psi_j\rangle$ in a list. To quickly find the index $j$ of the newly created state $|\psi_j\rangle$ in the Hilbert space, we utilize a lookup table which maps a state $|\psi\rangle$ to its index in the many-body basis. This is implemented via a hash-map (also referred to as dictionary) with lookup time $\mathcal{O}(1)$, allowing fast lookup at the expense of some memory to store the copy of the Hilbert space.

The second key ingredient of our implementation is the pre-computation of reduced single-particle matrix elements, which allows us to compute the set of transitions between states in the separate subspaces for the two particle species, i.e.,

$$|\tilde{\psi}_\sigma\rangle = \hat{\psi}_{\sigma,\alpha}^\dagger \hat{\psi}_{\sigma,\beta}|\psi_\sigma\rangle. \tag{41}$$

Note that $\alpha$ and $\beta$ denote indicies in the single-particle basis, i.e., the above expression describes a hop from a particle of spin $\sigma$ from orbital $\beta$ to orbital $\alpha$. All one-body terms are characterized by these "single-particle hops". Moreover, due to the spin-conserving nature of the considered particle interactions, the interaction term between $\uparrow$- and $\downarrow$ particles can be represented by the sum of all possible combinations of $\uparrow$- and $\downarrow$-hops. For interaction between spin-alike particles, some additional computational effort arises because these single-particle hops have to be re-computed in the inner loop (since the original state of the species changes and things like pair-hops come into play).

The collected lists of indicies $\{i\}$ and $\{j\}$ together with the matrix elements $h_{i,j}$ constitute a sparse representation of the full Hamiltonian matrix. With these ingredients an instance of the type `Hamiltonian` is created, which reflects an explicit convenience wrapper for sparse matrix objects. This object will later be passed to the diagonalization routines (based on ARPACK [21]) to obtain the lower portion of the spectrum.

# B  $\alpha$-coefficients for the 1D HO

In this appendix we present a brief derivation of the $\alpha$ coefficients that encode the transformation from the 'Cartesian representation' to the 'relative representation' in Eq. (12), see also Refs. [55, 56].

A suitable form for calculations can be obtained by using the identity $\mathbb{1} = \int \mathrm{d}x_1\mathrm{d}x_2\,|x_1x_2\rangle\langle x_1x_2|$,

which gives us the following equation for the coefficients $\alpha_{ij,ab}$

$$_R\langle\phi_a\phi_b|\phi_i\phi_j\rangle_C = \int dx_1 dx_2 \; _R\langle\phi_a\phi_b|x_1 x_2\rangle\langle x_1 x_2|\phi_i\phi_j\rangle_C \tag{42}$$

$$= \int dx_1 dx_2 \; \phi_a^*\left(\frac{x_1 - x_2}{\sqrt{2}}\right)\phi_b^*\left(\frac{x_1 + x_2}{\sqrt{2}}\right)\phi_i(x_1)\phi_j(x_2) \tag{43}$$

$$= \int dx_1 dx_2 \; \phi_a^*(x)\phi_b^*(X)\phi_i(x_1)\phi_j(x_2) \equiv \alpha_{ij,ab}, \tag{44}$$

which is nothing but Eq. (12). Note, that we could have also inserted $\mathbb{1} = \int dx dX \; |xX\rangle\langle xX|$ to the same effect. In the same way, we can write the backwards transformation

$$|\phi_a\phi_b\rangle_R = \sum_{i,j} |\phi_i\phi_j\rangle_C \; _C\langle\phi_i\phi_j|\phi_a\phi_b\rangle_R, \tag{45}$$

where

$$_C\langle\phi_i\phi_j|\phi_a\phi_b\rangle_R = \int dx_1 dx_2 \; _C\langle\phi_i\phi_j|x_1 x_2\rangle\langle x_1 x_2|\phi_a\phi_b\rangle_R \tag{46}$$

$$= \int dx_1 dx_2 \; \phi_i^*(x_1)\phi_j^*(x_2)\phi_a(x)\phi_b(X) \equiv \beta_{ab,ij}. \tag{47}$$

Note that it holds that

$$\alpha_{ij,ab} = \beta_{ab,ij}^*. \tag{48}$$

For the explicit form of $\alpha_{ij,ab}$ we may exploit some standard properties of Hermite polynomials, mainly

$$H_n(x+y) = 2^{-n/2}\sum_{k=0}^n \frac{n!}{k!(n-k)!}H_{n-k}(x\sqrt{2})H_k(y\sqrt{2}). \tag{49}$$

Using this identity we rewrite $\alpha$ (we measure now all distances in the harmonic oscillator units)

$$\alpha_{nk,a(n+k-a)} = \frac{\sqrt{a!(n+k-a)!}}{2^{3(n+k)/2}\pi\sqrt{n!k!}}\sum_{l=0}^a \frac{(-1)^l}{l!(a-l)!}\sum_{f=0}^{n+k-a}\frac{1}{f!(n+k-a-f)!}\gamma_{n,a-l,n+k-a-f}\gamma_{k,l,f}, \tag{50}$$

where

$$\gamma_{k,l,f} = \int_{-\infty}^{\infty} dy \; e^{-y^2}H_k(y)H_l(y)H_f(y). \tag{51}$$

The value of $\gamma$ may be found in [57]. Notice that if $k+l+f$ is odd then the integrand is an odd function, and $\gamma = 0$. This allows us to introduce $M = (k+l+f)/2$. $\gamma$ is also zero if $k > M$ or $l > M$ or $f > M$. For all other values of the parameters we have

$$\gamma_{k,l,f} = 2^M\sqrt{\pi}\frac{k!l!f!}{(M-k)!(M-l)!(M-f)!}. \tag{52}$$

Using this expression, we obtain

$$\alpha_{nk,a(n+k-a)} = \sqrt{\frac{n!k!a!(n+k-a)!}{2^{n+k}}}\sum_{l=0}^a (-1)^l Z_{nka}, \tag{53}$$

with

$$Z_{nka} = \sum_{f=0}^{n+k-a} \frac{1}{(M_1 - k)!(M_1 - l)!(M_1 - f)!(M_2 - n)!(M_2 - a + l)!(M_2 - n - k + a + f)!}.$$

Here, $M_1 = (k + l + f)/2$ and $M_2 = n + (k - l - f)/2$. Notice that $M_2 = n + k - M_1$. This observation means that $l = k - f$ to satisfy the condition that both $M_1 - k$ and $M_2 - n$ are positive. Therefore, we can further simplify our expression

$$\alpha_{nk,a(n+k-a)} = \sqrt{\frac{n!k!a!(n + k - a)!}{2^{n+k}}} \sum_{l=0}^{a} \frac{(-1)^l}{(k - l)!l!(n - a + l)!(a - l)!}, \tag{54}$$

if both $k - l$ and $n - a + l$ are positive. Otherwise the expression in the sum is zero. To check that these results are correct, we have compared the final result to a brute-force calculation of $\alpha$.

## C  Benchmark data

In this appendix, we provide benchmark data for $N = 3$ to $6$ trapped fermions in various configurations obtained with the bare interaction as well as the effective interaction (see the main text for more detail). For the three-body system we compare our results to the results of a variationally improved basis set from Ref. [58]. The results are summarized in Table 1. The values for the cutoff $n_b$ have been chosen such that the diagonalization completes in a few minutes for a regular personal machine. For completeness, the size of the corresponding many-body basis is also noted in the table.

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

| system size | method | dim $\mathcal{H}$ | $g = -4.0$ | $g = -1.0$ | $g = 1.0$ | $g = 4.0$ |
|---|---|---|---|---|---|---|
| $2\uparrow + 1\downarrow$ | bare ($n_b = 25$) | 7500 | 1.3320 | 1.8098 | 3.0141 | 3.8306 |
| | bare ($n_b = 50$) | 61250 | $-1.6476$ | 1.3637 | 3.0074 | 3.7919 |
| | effective ($n_b = 25$) | 7500 | $-2.4445$ | 1.7726 | 2.9949 | 3.7204 |
| | effective ($n_b = 35$) | 20825 | $-2.4821$ | 1.7716 | 2.9945 | 3.7186 |
| | variational method [58] | — | $-2.5865$ | 1.7685 | 2.9935 | 3.7140 |
| $3\uparrow + 1\downarrow$ | bare ($n_b = 15$) | 6825 | 1.0279 | 4.1830 | 5.6772 | 6.9455 |
| | bare ($n_b = 25$) | 57500 | 0.7397 | 4.1682 | 5.6666 | 6.8672 |
| | effective ($n_b = 15$) | 6825 | $-0.3420$ | 4.1280 | 5.6424 | 6.7017 |
| | effective ($n_b = 25$) | 57500 | $-0.4163$ | 4.1248 | 5.6407 | 6.6908 |
| $2\uparrow + 2\downarrow$ | bare ($n_b = 15$) | 11025 | $-2.6909$ | 2.7588 | 4.9649 | 6.6501 |
| | bare ($n_b = 20$) | 36100 | $-2.9921$ | 2.7450 | 4.9560 | 6.5880 |
| | effective ($n_b = 15$) | 11025 | $-5.3016$ | 2.6720 | 4.9169 | 6.3359 |
| | effective ($n_b = 20$) | 36100 | $-5.3507$ | 2.6690 | 4.9152 | 6.3267 |
| $3\uparrow + 2\downarrow$ | bare ($n_b = 12$) | 14520 | $-0.8874$ | 4.9623 | 7.8034 | 10.4114 |
| | bare ($n_b = 16$) | 67200 | $-1.2057$ | 4.9440 | 7.7882 | 10.2689 |
| | effective ($n_b = 12$) | 14520 | $-3.8076$ | 4.8485 | 7.7271 | 9.8185 |
| | effective ($n_b = 16$) | 67200 | $-3.8535$ | 4.8422 | 7.7239 | 9.7904 |
| $4\uparrow + 1\downarrow$ | bare ($n_b = 12$) | 5940 | 4.3009 | 7.5757 | 9.3166 | 11.0538 |
| | bare ($n_b = 18$) | 55080 | 4.0616 | 7.5609 | 9.3037 | 10.9256 |
| | effective ($n_b = 12$) | 5490 | 2.7601 | 7.5071 | 9.2676 | 10.6458 |
| | effective ($n_b = 18$) | 55080 | 2.7196 | 7.5043 | 9.2652 | 10.6233 |
| $3\uparrow + 3\downarrow$ | bare ($n_b = 10$) | 14400 | $-1.2519$ | 6.8545 | 10.8608 | 14.9295 |
| | bare ($n_b = 13$) | 81796 | $-1.6599$ | 6.8316 | 10.8470 | 14.6662 |
| | effective ($n_b = 10$) | 14400 | $-5.8959$ | 6.6748 | 10.7384 | 13.8307 |
| | effective ($n_b = 13$) | 81796 | $-5.8919$ | 6.6740 | 10.7371 | 13.8085 |
| $4\uparrow + 2\downarrow$ | bare ($n_b = 10$) | 9450 | 2.0311 | 8.2292 | 11.5899 | 15.1985 |
| | bare ($n_b = 13$) | 55770 | 1.7477 | 8.2111 | 11.5726 | 14.9701 |
| | effective ($n_b = 10$) | 9450 | $-1.2459$ | 8.0839 | 11.4835 | 14.2110 |
| | effective ($n_b = 13$) | 55770 | $-1.2284$ | 8.0832 | 11.4819 | 14.1830 |
| $5\uparrow + 1\downarrow$ | bare ($n_b = 12$) | 9504 | 8.4773 | 11.9745 | 13.9335 | 16.07656 |
| | bare ($n_b = 16$) | 69888 | 8.3066 | 11.9636 | 13.9231 | 15.9511 |
| | effective ($n_b = 12$) | 9504 | 6.8678 | 11.8990 | 13.8761 | 15.5305 |
| | effective ($n_b = 16$) | 69888 | 6.8534 | 11.8978 | 13.8749 | 15.5239 |

Table 1: The ground-state energy for systems with three to six particles for different interaction strengths $g$. Cutoffs have been chosen such that the calculations can be performed in a few minutes on a regular personal machine.

[6] J. Bjerlin, S. M. Reimann and G. M. Bruun, *Few-body precursor of the higgs mode in a fermi gas*, Phys. Rev. Lett. **116**, 155302 (2016), doi:10.1103/PhysRevLett.116.155302.

[7] N. L. Harshman, *Symmetries of three harmonically trapped particles in one dimension*, Phys. Rev. A **86**, 052122 (2012), doi:10.1103/PhysRevA.86.052122.

[8] P. Kościk, *Optimized configuration interaction approach for trapped multiparticle systems interacting via contact forces*, Phys. Lett. A **382**(36), 2561 (2018), doi:10.1016/j.physleta.2018.06.025.

[9] P. Jeszenszki, H. Luo, A. Alavi and J. Brand, *Accelerating the convergence of exact diagonalization with the transcorrelated method: Quantum gas in one dimension with contact interactions*, Phys. Rev. A **98**, 053627 (2018), doi:10.1103/PhysRevA.98.053627.

[10] P. Jeszenszki, U. Ebling, H. Luo, A. Alavi and J. Brand, *Eliminating the wave-function singularity for ultracold atoms by a similarity transformation*, Phys. Rev. Research **2**, 043270 (2020), doi:10.1103/PhysRevResearch.2.043270.

[11] I. Stetcu, B. R. Barrett, U. van Kolck and J. P. Vary, *Effective theory for trapped few-fermion systems*, Phys. Rev. A **76**, 063613 (2007), doi:10.1103/PhysRevA.76.063613.

[12] Y. Alhassid, G. F. Bertsch and L. Fang, *New effective interaction for the trapped fermi gas*, Phys. Rev. Lett. **100**, 230401 (2008), doi:10.1103/PhysRevLett.100.230401.

[13] J. Rotureau, *Interaction for the trapped fermi gas from a unitary transformation of the exact two-body spectrum*, Eur. Phys. J. D **67**(7), 153 (2013), doi:10.1140/epjd/e2013-40156-8.

[14] T. Ernst, D. W. Hallwood, J. Gulliksen, H.-D. Meyer and J. Brand, *Simulating strongly correlated multiparticle systems in a truncated hilbert space*, Phys. Rev. A **84**, 023623 (2011), doi:10.1103/PhysRevA.84.023623.

[15] M. Avakian, G. Pogosyan, A. Sissakian and V. Ter-Antonyan, *Spectroscopy of a singular linear oscillator*, Phys. Lett. A **124**(4), 233 (1987), doi:10.1016/0375-9601(87)90627-X.

[16] T. Busch, B.-G. Englert, K. Rzażewski and M. Wilkens, *Two cold atoms in a harmonic trap*, Found. Phys. **28**(4), 549 (1998), doi:10.1023/A:1018705520999.

[17] E. J. Lindgren, J. Rotureau, C. Forssén, A. G. Volosniev and N. T. Zinner, *Fermionization of two-component few-fermion systems in a one-dimensional harmonic trap*, New Journal of Physics **16**(6), 063003 (2014), doi:10.1088/1367-2630/16/6/063003.

[18] https://github.com/rammelmueller/FermiFCI.jl.

[19] https://github.com/rammelmueller/fermifci_data_repo.

[20] L. Rammelmüller, D. Huber, M. Čufar, J. Brand, H.-W. Hammer and A. G. Volosniev, *Magnetic impurity in a one-dimensional few-fermion system*, doi:10.48550/ARXIV.2204.01606 (2022).

[21] R. B. Lehoucq, D. C. Sorensen and C. Yang, *ARPACK Users' Guide*, Society for Industrial and Applied Mathematics, doi:10.1137/1.9780898719628 (1998).

[22] F. Serwane, G. Zürn, T. Lompe, T. B. Ottenstein, A. N. Wenz and S. Jochim, *Deterministic Preparation of a Tunable Few-Fermion System*, Science **332**(6027), 336 (2011), doi:https://doi.org/10.1126/science.1201351.

[23] G. Zürn, A. N. Wenz, S. Murmann, A. Bergschneider, T. Lompe and S. Jochim, *Pairing in Few-Fermion Systems with Attractive Interactions*, Phys. Rev. Lett. **111**(17), 175302 (2013), doi:10.1103/PhysRevLett.111.175302.

[24] S. Murmann, F. Deuretzbacher, G. Zürn, J. Bjerlin, S. Reimann, L. Santos, T. Lompe and S. Jochim, *Antiferromagnetic Heisenberg Spin Chain of a Few Cold Atoms in a One-Dimensional Trap*, Phys. Rev. Lett. **115**(21), 215301 (2015), doi:10.1103/PhysRevLett.115.215301.

[25] S. E. Gharashi and D. Blume, *Correlations of the upper branch of 1d harmonically trapped two-component fermi gases*, Phys. Rev. Lett. **111**, 045302 (2013), doi:10.1103/PhysRevLett.111.045302.

[26] T. Grining, M. Tomza, M. Lesiuk, M. Przybytek, M. Musiał, P. Massignan, M. Lewenstein and R. Moszynski, *Many interacting fermions in a one-dimensional harmonic trap: a quantum-chemical treatment*, New J. Phys. **17**(11), 115001 (2015), doi:10.1088/1367-2630/17/11/115001.

[27] M. Moshinsky, *The Harmonic Oscillator in Modern Physics: From Atoms to Quarks*, Gordon and Breach (1969).

[28] *Mathematica, Version 13.1*, Wolfram Research Inc., Champaign, IL, 2022.

[29] F. Deuretzbacher, D. Becker, J. Bjerlin, S. M. Reimann and L. Santos, *Quantum magnetism without lattices in strongly interacting one-dimensional spinor gases*, Phys. Rev. A **90**, 013611 (2014), doi:10.1103/PhysRevA.90.013611.

[30] A. G. Volosniev, D. V. Fedorov, A. S. Jensen, M. Valiente and N. T. Zinner, *Strongly interacting confined quantum systems in one dimension*, Nature Commun **5**, 5300 (2014), doi:10.1038/ncomms6300.

[31] D. Pęcak and T. Sowiński, *Few strongly interacting ultracold fermions in one-dimensional traps of different shapes*, Phys. Rev. A **94**, 042118 (2016), doi:10.1103/PhysRevA.94.042118.

[32] A. Volosniev, *Strongly Interacting One-dimensional Systems with Small Mass Imbalance*, Few-Body Systems **58**, 54 (2017), doi:10.1007/s00601-017-1227-0.

[33] N. T. Zinner, K. Mølmer, C. Özen, D. J. Dean and K. Langanke, *Shell-model monte carlo simulations of the bcs-bec crossover in few-fermion systems*, Phys. Rev. A **80**, 013613 (2009), doi:10.1103/PhysRevA.80.013613.

[34] M. Rontani, G. Eriksson, S. Åberg and S. M. Reimann, *On the renormalization of contact interactions for the configuration-interaction method in two-dimensions*, J. Phys. B **50**(6), 065301 (2017), doi:10.1088/1361-6455/aa606a.

[35] G. Bougas, S. I. Mistakidis and P. Schmelcher, *Analytical treatment of the interaction quench dynamics of two bosons in a two-dimensional harmonic trap*, Phys. Rev. A **100**, 053602 (2019), doi:10.1103/PhysRevA.100.053602.

[36] M. Lüscher, *Volume dependence of the energy spectrum in massive quantum field theories. i. stable particle states*, Comm. Math. Phys. **104**(2), 177 (1986).

[37] M. Lüscher, *Volume dependence of the energy spectrum in massive quantum field theories. ii. scattering states*, Comm. Math. Phys. **105**(2), 153 (1986), doi:10.1007/BF01211589.

[38] K. Suzuki and S. Y. Lee, *Convergent Theory for Effective Interaction in Nuclei\**, Prog. Theor. Phys. **64**(6), 2091 (1980), doi:10.1143/PTP.64.2091.

[39] P. Navrátil, J. P. Vary and B. R. Barrett, *Properties of $^{12}c$ in the ab initio nuclear shell model*, Phys. Rev. Lett. **84**, 5728 (2000), doi:10.1103/PhysRevLett.84.5728.

[40] A. F. Lisetskiy, B. R. Barrett, M. K. G. Kruse, P. Navratil, I. Stetcu and J. P. Vary, *Ab-initio shell model with a core*, Phys. Rev. C **78**, 044302 (2008), doi:10.1103/PhysRevC.78.044302.

[41] A. Dehkharghani, A. Volosniev, J. Lindgren, J. Rotureau, C. Forssén, D. Fedorov, A. Jensen and N. Zinner, *Quantum magnetism in strongly interacting one-dimensional spinor bose systems*, Scientific Reports **5**(1), 10675 (2015), doi:10.1038/srep10675.

[42] A. G. Volosniev and H.-W. Hammer, *Flow equations for cold bose gases*, New J. Phys. **19**(11), 113051 (2017), doi:10.1088/1367-2630/aa9011.

[43] T. Grining, M. Tomza, M. Lesiuk, M. Przybytek, M. Musiał, R. Moszynski, M. Lewenstein and P. Massignan, *Crossover between few and many fermions in a harmonic trap*, Phys. Rev. A **92**, 061601 (2015), doi:10.1103/PhysRevA.92.061601.

[44] A. N. Wenz, G. Zürn, S. Murmann, I. Brouzos, T. Lompe and S. Jochim, *From few to many: Observing the formation of a fermi sea one atom at a time*, Science **342**(6157), 457 (2013), doi:10.1126/science.1240516.

[45] F. Chevy and C. Mora, *Ultra-cold polarized fermi gases*, Rep. Prog. Phys. **73**(11), 112401 (2010), doi:10.1088/0034-4885/73/11/112401.

[46] P. Massignan, M. Zaccanti and G. M. Bruun, *Polarons, dressed molecules and itinerant ferromagnetism in ultracold fermi gases*, Rep. Prog. Phys. **77**(3), 034401 (2014), doi:10.1088/0034-4885/77/3/034401.

[47] P. D'Amico and M. Rontani, *Pairing of a few fermi atoms in one dimension*, Phys. Rev. A **91**, 043610 (2015), doi:10.1103/PhysRevA.91.043610.

[48] L. Bayha, M. Holten, R. Klemt, K. Subramanian, J. Bjerlin, S. M. Reimann, G. M. Bruun, P. M. Preiss and S. Jochim, *Observing the emergence of a quantum phase transition shell by shell*, Nature **587**(7835), 583 (2020), doi:10.1038/s41586-020-2936-y.

[49] M. Holten, L. Bayha, K. Subramanian, S. Brandstetter, C. Heintze, P. Lunt, P. M. Preiss and S. Jochim, *Observation of cooper pairs in a mesoscopic 2d fermi gas* (2021), `2109.11511`.

[50] G. H. Booth, A. J. W. Thom and A. Alavi, *Fermion monte carlo without fixed nodes: A game of life, death, and annihilation in slater determinant space*, J. Chem. Phys. **131**(5), 054106 (2009), doi:10.1063/1.3193710.

[51] C. E. Berger, L. Rammelmüller, A. C. Loheac, F. Ehmann, J. Braun and J. E. Drut, *Complex Langevin and other approaches to the sign problem in quantum many-body physics*, Phys. Rept. **892**, 1 (2021), doi:10.1016/j.physrep.2020.09.002.

[52] L. Cao, V. Bolsinger, S. I. Mistakidis, G. M. Koutentakis, S. Krönke, J. M. Schurer and P. Schmelcher, *A unified ab initio approach to the correlated quantum dynamics of ultracold fermionic and bosonic mixtures*, J. Chem. Phys. **147**(4), 044106 (2017), doi:10.1063/1.4993512.

[53] A. U. J. Lode, C. Lévêque, L. B. Madsen, A. I. Streltsov and O. E. Alon, *Colloquium: Multiconfigurational time-dependent hartree approaches for indistinguishable particles*, Rev. Mod. Phys. **92**, 011001 (2020), doi:10.1103/RevModPhys.92.011001.

[54] Y. Gwak, O. V. Marchukov and U. R. Fischer, *Benchmarking the multiconfigurational hartree method by the exact wavefunction of two harmonically trapped bosons with contact interaction*, Ann. Phys. **434**, 168592 (2021), doi:10.1016/j.aop.2021.168592.

[55] Y. Smirnov, *Talmi transformation for particles with different masses (ii)*, Nuclear Physics **39**, 346 (1962), doi:https://doi.org/10.1016/0029-5582(62)90398-X.

[56] L. M. Robledo, *Separable approximation to two-body matrix elements*, Phys. Rev. C **81**, 044312 (2010), doi:10.1103/PhysRevC.81.044312.

[57] R. D. Lord, *Some Integrals Involving Hermite Polynomials*, J. London Math. Soc. **s1-24**(2), 101 (1949), doi:10.1112/jlms/s1-24.2.101.

[58] P. D'Amico and M. Rontani, *Three interacting atoms in a one-dimensional trap: a benchmark system for computational approaches*, J. Phys. B **47**(6), 065303 (2014), doi:10.1088/0953-4075/47/6/065303.