# Peer review of "A modular implementation of an effective interaction approach for harmonically trapped fermions in 1D"

_SciPost Physics Codebases, doi:SciPost Phys. Codebases 12-r1.0 (2023) , SciPost Phys. Codebases 12 (2023)_

## Round 1 · Referee Report · Tim Keller (Referee 1) · 2022-4-7

Strengths

1 - Easy to use and well documented code with versatile applications
2 - Providing many examples of wide variety and complexity
3 - Accompanying paper is well structured and provides a thorough introduction to the problem the code is solving

Weaknesses

1 - Some issues regarding implementation, reproducibility and interpretation of the examples
2 - Actual technical implementation of constructing the Hamiltonian is not described

Report

In the present paper the authors introduce a codebase providing a generic exact diagonalization scheme for two-component systems covering arbitrary interactions that can be described via two-body Hamiltonians. Its main goal is to provide a basic framework to build upon, both for newcomers to the field as well as for trying out new optimizations for experts. The provided examples and use cases are focused on harmonically trapped few-fermion systems in one dimension, particularly on improving the achievable precision via Hilbert space truncations and via an effective interaction approach. Both the code and the provided examples/benchmarks are well written and well documented, including installation guidelines and mentioning required external packages, while adhering to high-level programming standards and conventions. The authors do not specifically mention already existing software, but to the best of my knowledge this is the first publicly available implementation of an exact diagonalization scheme in the context of ultracold atoms which also specifically includes the effective interaction approach.

Overall, the submission fulfills the acceptance criteria and therefore I recommend publication after the points raised in the following as well as the specifically requested changes are addressed.

In Section 2.3.2, the specific technical implementation of how the Hamiltonian is constructed is not explicitly described, even though it is the package's core functionality (next to the diagonalization part which is handled by existing implementations/external packages). The authors have chosen a non-trivial and very elegant solution involving lookup tables and pre-computing the indices of the non-zero matrix elements via allowed state changes, which I believe deserves to be explained in the main text.

The provided examples offer a nice variety of use cases of increasing complexity. However, I believe that their presentation and implementation is not ideal and could be improved.

Firstly, while the correct data files to reproduce the figures shown in the paper are provided, in the current version the example code does not reproduce this data without adjusting essentially all parameters regarding system size, number of basis states and eigenvalues, range of the coupling parameter and so on (except for Example III). Or as in Example IV they do not even output the correct data format that is needed to produce the plots shown in the paper. Admittedly, this forced interaction with the code provides a good learning experience but the examples should be fixed to at least reproduce the paper's figures 'out-of-the-box' without needing further adjustments. Secondly, I do not understand the reasoning behind separating the examples into a .jl file producing the data and into a Python notebook producing the plots. I believe it would increase the educational value of the examples if both data production and plotting were combined, e.g. into one coherent Julia notebook for each plot or for each example and with the (very detailed and well written!) documentation turned into markup text.

It might also be beneficial to incorporate code like tensor_construction.jl or the single-particle basis state representations into the main package for convenience. I can understand the reasoning behind keeping the FermiFCI package as simple as possible, as demonstrated in Fig. 1, but seeing that it is necessary to manually include these files in every example presented here and that harmonically trapped particles make up a large amount of possible use cases, it seems like a strong argument for incorporating them into the main package. This would also simplify the slightly convoluted file structure in the example repository.

I would also like to suggest the authors to include the implementations for the energy-cutoff as well as the effective interaction approach into the main package. After all, the submission is titled 'A modular implementation of an effective interaction approach for harmonically trapped fermions in 1D'. This is not a requested change since in both cases the code is already provided in the example package and can be copied if necessary. However, I believe that on one hand it would allow to simplify the examples even more for newcomers and on the other hand it would provide a known benchmark for people wanting to use the package as a stepping stone to try out new approximations beyond these established techniques.

Regarding Example I, I do not fully understand the emergence of the apparent additional energy level in Fig. 2 compared to the benchmark data and how, according to the authors, this relates to discarded basis states. Particularly since the additional energy level seems to agree with the highest energy level (5th lowest state) shown in Ref. [25] (but not shown in Fig. 2!) once it crosses the 4th lowest state roughly for values $g>5$. So the discrepancy only seems to occur for couplings $g<5$. I would like the authors to provide a more detailed explanation of this discrepancy and the apparent level crossings that do not occur in the benchmark data. Judging from the data shown in Table 1 of Appendix B, it seems that, at least for attractive interactions, the plain-cutoff in combination with the bare interaction results in a very large error that might cause a higher lying level to appear as the 'additional' level below certain coupling strengths. In that case simply restricting the example to $g>5$ or to repulsive couplings might be less confusing. This also raises the additional question why panel A does not show all six levels of the ground state manifold that reach the same energy level in the limit $g\rightarrow\infty$ (as indicated in the caption) similar to how panel B shows all four levels in the case of 3 spin-up and 1 spin-down particle.

Regarding Figure 5, clearly the discrepancy in the density can be reduced by using more basis states but I am curious if there is an intuitive explanation as to why using the energy-cutoff with $n_b = 24$ actually produces what appears to be a small dip around $x=0$? Naively one would expect that the lower energy basis states are mainly contributing to the central region and should therefore produce a quite good agreement in that area while higher energy basis states might be necessary to resolve finer details in the density.

Overall, there is a noticeable amount of typos, both in the paper and also in the code in the form of apparently copy and pasted comments (e.g. same description for create() and annihilate() functions, all examples are called run_exII.jl in the header,...) to the point where I feel it is necessary to mention it and there are some phrases, particularly 'cheap numerical effort' above Eq. (34), that might have unintended negative connotations. Also I recommend to replace potentially misleading expressions like 'for the 3D harmonic oscillator' in section 4.1 and 'two-body interaction for the 1D harmonic oscillator' in section 4.3 by something like 'for harmonically trapped Fermions in one (three) dimension(s)' respectively.

Finally, I would like to ask the authors to comment on the following points and maybe even incorporate their answers into the paper if appropriate. 1) In the provided examples only the case of diagonal one-body matrix terms is covered, but the code seems to be able to also handle the non-diagonal case (as is also stated below Eq. (6)). What would be the general approach in such a case? Expressing the non-diagonal term in a 'simple' basis like the harmonic oscillator eigenstates where also the interaction coefficients are known or is it advantageous to first diagonalize the one-body terms and then work in the (possibly complicated) diagonal basis?

2) Similarly, the code is presented in the context of a two-component fermionic system where only the inter-component interaction is relevant but the 'UP-UP' and 'DOWN-DOWN' terms are also implemented in construction.jl. Does this mean that the code can also be used for exact diagonalization of a bosonic system if the basis states are adjusted accordingly?

Requested changes

1 - Below Eq. (1), please explicitly write down the fermionic anticommutation relations for the sake of clarity.

2 - Maybe I'm missing the reasoning, but 'flavor' seems like an odd choice to describe different types of Fermions in an ultracold atoms setting, especially since throughout the paper only two types, spin-up and spin-down, are distinguished.

3 - In Section 2.3.2, please expand on the 'inner workings' of the Hamiltonian construction as indicated in the report above.

4 - The interaction coefficients presented in Section 3.1.1 and Appendix A for the harmonic oscillator basis are known in the literature as Talmi-Brody-Moshinsky coefficients. Please add some references, e.g. L. Robledo, Phys. Rev. C 81, 044312 (2010).

5 - Please remove the erroneous factor $g$ in Eq. (11) with the general two-body potential.

6 - There seems to be a change in index notation from $nk\rightarrow ij$ within Eq. (12)/(13). Please correct or clarify it.

7 - Please clarify the issues raised in the report above regarding the seemingly additional energy level in Figure 2 / Example I.

8 - Is there a reason why the diagram in Fig. 3 shows a setup of 3 spin-up and 2 spin-down particles in contrast to the 3 spin-up and 1 spin-down setup in the accompanying table? If not, please adjust it.

9 - In the caption of Fig. 6 and in the main text at the bottom of page 15, the number of spin-up and spin-down components is switched compared to the data shown in the figure.

10 - Just before Section 4, please replace 'if particles separate' by 'if the two species separate' to be more precise.

11 - In Eq. (25), the indices $na$ of the first term seem to be wrong. Please swap them to $an$ or exchange the order of the indices and the Hermitian transpose operator.

12 - Please fix Eq. (31) by enclosing the argument of the Gamma function in the denominator in parentheses. Also it is missing a factor of $\sqrt{2}$ if the earlier definitions of $g$ for the coupling and $r = (x_1 - x_2)/\sqrt{2}$ for the relative coordinate are being used.

13 - I believe Eq. (33) needs to be multiplied by $(-1)$ for the correct expression, even though it doesn't affect the transformation as you acknowledge in the comments of eff_relative.jl

14 - In the caption of Fig. 7, please explicitly indicate the value of $g=5$ that the inset of panel A is showing.

15 - Please add the missing spin-down symbol in the paragraph above Eq. (34).

16 - The right-hand side of Eq. (37) should have a minus instead of a plus in front of the $\Delta(x)$ term to be consistent with Eq. (36)

17 - I cannot see how Eq. (38) is obtained from the expression in the text. Please provide some intermediate steps in the calculation.

18 - In Fig. 8, please explicitly indicate the system configuration of 3 spin-up particles and 1 spin-down particle in the figure and/or caption.

19 - The program was only able to load the provided HDF5 file with the precomputed alpha-coefficients after adjusting the file path to abspath(joinpath(@DIR,"../alpha_coefficients_ho1d.hdf5")), i.e. providing an absolute path. This seems to be an issue with the HDF5 package on macOS since including .jl files via relative paths worked fine, however it might be worth keeping an eye on platform independence regarding file paths.

20 - Please improve the provided examples based on the issues raised in the report regarding reproducibility and user-friendliness.

  • validity: high
  • significance: high
  • originality: high
  • clarity: top
  • formatting: excellent
  • grammar: good

Author:  Lukas Rammelmüller  on 2022-11-05  [id 2986]

(in reply to Report 1 by Tim Keller on 2022-04-07)

We thank the Referee for taking the time to review our work. We very much appreciate the overall positive report and constructive criticism that helped us to improve the presentation of our work. The revised version of the report implements the requested changes as discussed below. For convenience, we shall also quote the comments of the Referee.

  1. Referee: Below Eq. (1), please explicitly write down the fermionic anticommutation relations for the sake of clarity. Our reply: We have implemented this comment, please see the revised version.

  2. Referee: Maybe I'm missing the reasoning, but 'flavor' seems like an odd choice to describe different types of Fermions in an ultracold atoms setting, especially since throughout the paper only two types, spin-up and spin-down, are distinguished. Our reply: The word “flavour” is indeed odd but it is commonly used in the context o few-body one-dimensional systems, see, e.g., New J. Phys. 18 013030 (2016) and Phys. Rev. A 97, 033602 (2018). After meditating on the Referee’s comment, we decided to leave the text as is. The word ‘flavor’ enters in the implementation of the code, therefore, changing a convention may introduce typos, which can be a source of confusion.

  3. Referee: In Section 2.3.2, please expand on the 'inner workings' of the Hamiltonian construction as indicated in the report above. Our reploy: We thank the Referee for this comment. We have included Appendix A that discusses how the code constructs the Hamiltonian.

  4. Referee: The interaction coefficients presented in Section 3.1.1 and Appendix A for the harmonic oscillator basis are known in the literature as Talmi-Brody-Moshinsky coefficients. Please add some references, e.g. L. Robledo, Phys. Rev. C 81, 044312 (2010). Our reply: We thank the Referee for referring us to the relevant literature. We were not aware of it. In the revised version we added a few references including the one suggested by the Referee.

  5. Referee: Please remove the erroneous factor g in Eq. (11) with the general two-body potential. Our reply: We have removed the factor in the revised version. We thank the Referee for noticing this typo.

  6. Referee: There seems to be a change in index notation from nk→ij within Eq. (12)/(13). Please correct or clarify it. Our reply: We have corrected this typo in the revised version. We thank the Referee for identifying it.

  7. Referee: Please clarify the issues raised in the report above regarding the seemingly additional energy level in Figure 2 / Example I. Our reply: We thank the Referee for highlighting this issue. Note that at g=0 the first excited state should be always double degenerate (one can either excite one spin-up or one spin-down particle). Our results feature this double degeneracy, but the benchmark results do not. Therefore, Ref. [25] has indeed discarded certain energy states. Most likely, these states are simply center-of-mass excitations that can be projected out in a harmonic trap. We have investigated this issue further and observed that our code can indeed reproduce the spectrum of Ref. [25] if we add a routine that eliminates the center-off-mass excitations. We have modified the text of the manuscript to address this issue and other comments from the Referee’s report.

  8. Referee: Is there a reason why the diagram in Fig. 3 shows a setup of 3 spin-up and 2 spin-down particles in contrast to the 3 spin-up and 1 spin-down setup in the accompanying table? If not, please adjust it. Our reply: Since the table contains information for both 3+1 and 2+2, we decided to sketch a set-up which is not related to these two systems. We hoped that this will help to avoid any confusion.

  9. Referee: In the caption of Fig. 6 and in the main text at the bottom of page 15, the number of spin-up and spin-down components is switched compared to the data shown in the figure. Our reply: We have corrected this typo in the revised version. We thank the Referee for identifying it.

  10. Referee: Just before Section 4, please replace 'if particles separate' by 'if the two species separate' to be more precise. Our reply: We have followed the recommendation of the Referee and changed the text.

  11. Referee: In Eq. (25), the indices na of the first term seem to be wrong. Please swap them to an or exchange the order of the indices and the Hermitian transpose operator. Our reply: We thank the Referee for identifying this notation as a potential source of confusion. We modified the equation accordingly.

  12. Referee: Please fix Eq. (31) by enclosing the argument of the Gamma function in the denominator in parentheses. Also it is missing a factor of √2 if the earlier definitions of g for the coupling and r=(x1−x2)/√2 for the relative coordinate are being used. Our reply: We thank the Referee for identifying this typo. We corrected it in the revised version.

  13. Referee: I believe Eq. (33) needs to be multiplied by (−1) for the correct expression, even though it doesn't affect the transformation as you acknowledge in the comments of eff_relative.jl Our reply: We have corrected the typo. We thank the Referee for identifying it.

  14. Referee: In the caption of Fig. 7, please explicitly indicate the value of g=5 that the inset of panel A is showing. Our reply: Following the recommendation of the Referee, we have added this information to the caption of the figure.

  15. Referee: Please add the missing spin-down symbol in the paragraph above Eq. (34). Our reply: We have added the symbol. We thank the Referee for pointing this typo out.

  16. Referee: The right-hand side of Eq. (37) should have a minus instead of a plus in front of the Δ(x) term to be consistent with Eq. (36) Our reply: We have modified the equation accordingly. We thank the Referee for identifying the typo.

  17. Referee: I cannot see how Eq. (38) is obtained from the expression in the text. Please provide some intermediate steps in the calculation. Our reply: Following the recommendation of the Referee, we have extended the discussion around Eq. (38). Equation (38) can be obtained using the Hermite numbers.

  18. Referee: In Fig. 8, please explicitly indicate the system configuration of 3 spin-up particles and 1 spin-down particle in the figure and/or caption. Our reply: We have modified the caption according to the recommendation of the Referee

  19. Referee: The program was only able to load the provided HDF5 file with the precomputed alpha-coefficients after adjusting the file path to abspath(joinpath(@DIR,"../alpha_coefficients_ho1d.hdf5")), i.e. providing an absolute path. This seems to be an issue with the HDF5 package on macOS since including .jl files via relative paths worked fine, however it might be worth keeping an eye on platform independence regarding file paths. Our reply: We thank the Referee for pointing out this potential stumbling block. We have added a “Troubleshooting” section to the README file of the package and mentioned this scenario.

  20. Referee: Please improve the provided examples based on the issues raised in the report regarding reproducibility and user-friendliness. Our reply: We thank the Referee for carefully considering the structure of our package as well as the associated data/script repository for this paper. In general, we agree with the Referee that a complete package with implemented single-particle levels for various standard problems as well as other related functions would likely be more convenient for the user. However, it was a design decision to not clutter the core package with problem-specific code in favor to keep the complete FermiFCI.jl package problem agnostic. The published code for the specific problems (harmonic oscillator, flat box, effective interaction, energy cutoff, tensor construction, ...) constitutes our own implementation for those problems using the core package – the goal is to demonstrate how a user would interact with FermiFCI.jl. We intend to keep this basic structure, however, in order to better connect to specific applications we have included a list (which will hopefully be extended in the future) of application of our package in the README file.

Regarding the production for plots: The decision to exclude a plot generator from the package was met for various reasons. Firstly, the authors firmly believe that handling data aggregation and post-processing (e.g., plots, extrapolations, comparison to literature, …) should considered as separate tasks not mixed into a single, potentially large and messy, notebook. Secondly, data production might run for quite some time for larger basis cutoffs, which would make the use of Julia notebooks cumbersome as they potentially have to be run on remote servers as opposed to the personal notebook. Simple data-collection scripts generalize easier to this use case. Lastly, the choice for using Python rather than Julia for plotting the data is purely based on the author’s preference (and knowledge) of the Python plotting features (as opposed to Julia).

Finally we would like to comment on ready-to-roll data collection scripts. Similarly to the argument above, data collection scripts that produce all data in a single run might not be the ideal use case, in particular once a scheduling system (like SLURM) comes into play. Various runs might have vastly different CPU and memory requirements, which are then difficult to bundle into a single script. It is more convenient to call the script with different parameters – or, more ideally, to use a configuration file altogether. The implementation here was of the simple type. However, to help the user with re-producing our data we have added relevant parameter sets (where easily possible) as comments.

Additional comments from the reply: Referee: Regarding Figure 5, clearly the discrepancy in the density can be reduced by using more basis states but I am curious if there is an intuitive explanation as to why using the energy-cutoff with nb=24 actually produces what appears to be a small dip around x=0? Naively one would expect that the lower energy basis states are mainly contributing to the central region and should therefore produce a quite good agreement in that area while higher energy basis states might be necessary to resolve finer details in the density. Our reply: According to our understanding, truncated Hilbert spaces overestimate the contribution of the potential energy. Therefore, the system interacts stronger (effectively). In other words, there is a somewhat higher ‘pressure’ inside of the system in a truncated Hilbert space in comparison to that in a full space. This increases the size of the system, and leads to the behavior presented in the figure.

Referee: Overall, there is a noticeable amount of typos, both in the paper and also in the code in the form of apparently copy and pasted comments (e.g. same description for create() and annihilate() functions, all examples are called run_exII.jl in the header,...) to the point where I feel it is necessary to mention it and there are some phrases, particularly 'cheap numerical effort' above Eq. (34), that might have unintended negative connotations. Also I recommend to replace potentially misleading expressions like 'for the 3D harmonic oscillator' in section 4.1 and 'two-body interaction for the 1D harmonic oscillator' in section 4.3 by something like 'for harmonically trapped Fermions in one (three) dimension(s)' respectively. Our reply: We thank the Referee for pointing out typos. We have corrected them as well as others that we found upon re-reading the manuscript.

Referee: Finally, I would like to ask the authors to comment on the following points and maybe even incorporate their answers into the paper if appropriate. 1) In the provided examples only the case of diagonal one-body matrix terms is covered, but the code seems to be able to also handle the non-diagonal case (as is also stated below Eq. (6)). What would be the general approach in such a case? Expressing the non-diagonal term in a 'simple' basis like the harmonic oscillator eigenstates where also the interaction coefficients are known or is it advantageous to first diagonalize the one-body terms and then work in the (possibly complicated) diagonal basis? Our reply: We thank the Referee for this comment. Yes, our code can be used to consider general one-body terms. Actually, we used FermiFCI in our recent preprint (arXiv:2204.01606). We have modified the text of the manuscript accordingly. Regarding a general approach to such a case: The answer depends strongly on the problem at hand and (unfortunately) a general answer cannot be given here. From a computational viewpoint, however, it is better to work in the basis that eventually results in a sparser Hamiltonian matrix, which reduces memory and often computational requirements.

Referee: Similarly, the code is presented in the context of a two-component fermionic system where only the inter-component interaction is relevant but the 'UP-UP' and 'DOWN-DOWN' terms are also implemented in construction.jl. Does this mean that the code can also be used for exact diagonalization of a bosonic system if the basis states are adjusted accordingly? Our reply: In principle one could also diagonalize bosonic systems – as the Referee points out, this can be achieved by considering intra-species couplings. In addition, the sign of the commutation needs to be disregarded, which can simply be done before multiplying the Hamiltonian with the appropriate coefficients. An appropriate line was added to the Outlook section of the manuscript.

---

## Round 1 · Referee Report · Anonymous (Referee 2) · 2022-4-13

Strengths

  1. The code is easy to use and well documented
  2. Several benchmarks are given, which also can be used as examples of how to use the code.
  3. The paper is clear and well written.

Weaknesses

  1. The added value of this specific implementation, as compared to existing software , is not clear.
  2. None of the benchmarks is against (semi-)analytical results.
  3. The physical interpretation and analysis of the given numerical results is not full.
  4. Performance vs. clearance; see below.

Report

The authors introduce a generic implementation of an exact diagonalization method, which can be used for numerical study of few-fermion systems.

The manuscript is well written, it contains detailed benchmarks against available numerical and experimental results, and the code is well documented.

However, before I could recommended for publication, I would like to understand better few points and offer some changes.

  1. One of the acceptance criteria for publishing in the journal is "a demonstrable need for the scientific community", and the user-guide should "highlight its added value as compared to existing software". Since several open-source exact diagonilization codes are available, the authors should describe better the uniqueness of their code.

  2. The authors prefer code clearance to its performance, which is a good practice in general. However, choosing to represent the states as specific types instead of bitsring, and to store the whole Hamiltonian matrix instead of writing function for matrix and vector multiplication, might limit the code applicability for larger systems. It might be useful to give also the other options in this two cases, or at least to estimate the computational cost of the exist implementation and describe how the user can change it.

Requested changes

  1. The authors benchmark their code against available numerical results, mostly from Grining et al., Ref. [25]. It might be nice to have also benchmark against (semi-)analytical results, for example those of Busch et al., Ref [16].

  2. The authors numerical results fit nicely those of Ref. [25]. However, in the attractive (g<0) part of Fig. 2 there is some deviation, which should be discussed. In addition, it seems that four lowest states are given for this code but only three for Ref. [25]. What is the reason for that?

  3. The numerical results should be better explained ; for example, in Fig. 7B the bare interaction results converge to the wrong results; why? What is the physical interpretation of the zigzag structure in Fig. 7D?

  4. An appropriate reference is needed for Mathematica (pp. 11).

  5. The code documentation should be completed. For example, in the readme file of the HO example it written - "TODO: describe the implementation and what happened here."

  • validity: top
  • significance: ok
  • originality: good
  • clarity: high
  • formatting: excellent
  • grammar: perfect

Author:  Lukas Rammelmüller  on 2022-11-05  [id 2987]

(in reply to Report 2 on 2022-04-13)

  1. Referee: The authors benchmark their code against available numerical results, mostly from Grining et al., Ref. [25]. It might be nice to have also benchmark against (semi-)analytical results, for example those of Busch et al., Ref [16]. Our reply: Please note that we use the (semi-)analytical results by Busch et al. to produce the effective interaction. Therefore, our energies are immediately converged for a two-body system when using the effective interaction. To avoid any possible misinterpretation of our results, we decided to leave any comparison to the (semi-)analytical results. Of course, if the Referee finds that such a comparison is still needed, we will be glad to include it.

  2. Referee: The authors numerical results fit nicely those of Ref. [25]. However, in the attractive (g<0) part of Fig. 2 there is some deviation, which should be discussed. In addition, it seems that four lowest states are given for this code but only three for Ref. [25]. What is the reason for that? Our reply: We thank the Referee for highlighting these issues. The deviation for g<0 is likely an artifact of the way we digitize the data from the figure of Ref. [25]. Unfortunately, it was impossible to obtain the data in the other ways. Regarding an additional level: note that at g=0 the first excited state should be always double degenerate (one can either excite one spin-up or one spin-down particle). Our results feature this double degeneracy, but the benchmark results do not. Therefore, Ref. [25] has discarded certain energy states. We have investigated this issue further and observed that our code can indeed reproduce the spectrum of Ref. [25] if we add a routine that eliminates the center-off-mass excitations. We have modified the text of the manuscript to explain this.

  3. Referee: The numerical results should be better explained ; for example, in Fig. 7B the bare interaction results converge to the wrong results; why? What is the physical interpretation of the zigzag structure in Fig. 7D? Our reply: In the revised version, we clarify that Fig. 7B shows the data obtained with bare interaction and effective interaction only for n_b=14. These data sets do now show results in the limit n_b\to\infty. The purpose of the Figure is to show that the data based upon the effective interaction is very close to the limit n_b\to\infty already for relatively small Hilbert spaces, whereas the results based upon the bare interaction are far off. Regarding the zigzag structure presented in Fig. 7D – this structure is an even-odd effect associated with BCS-type pairing. We have modified the text to explain these numerical results better.

  4. Referee: An appropriate reference is needed for Mathematica (pp. 11). Our reply: Following the recommendation of the Referee, we have added a reference for Mathematica.

  5. Referee: The code documentation should be completed. For example, in the readme file of the HO example it written - "TODO: describe the implementation and what happened here." Our reply: We thank the Referee for pointing out this shortcoming. We have added some information in the README as well as a link to the paper itself, which functions as a manual for the provided package.

---

## Round 2 · Referee Report · Anonymous (Referee 2) · 2022-11-22

Report

The authors responded well to most of my previous specific comments.

My general comments, however, were not addressed, and I would like to know the authors response.

Anyway, I recommended the current version for publication.

---

## Round 2 · Referee Report · Tim Keller (Referee 1) · 2022-11-25

Report

The authors have appropriately addressed all requested changes as well as the general comments and therefore I recommend publication of the current version.

---

## Editorial Decision

published